

# Mn/Ca intra-test variability in the benthic foraminifer
## *Ammonia tepida*

Jassin Petersen[1], Christine Barras[1], Antoine Bézos[1], Carole La[1], Lennart J. de Nooijer[2], Filip J.R. Meysman[3,4], Aurélia Mouret[1], Caroline P. Slomp[5], and Frans J. Jorissen[1]

[1]LPG UMR CNRS 6112, University of Nantes, University of Angers, UFR Sciences, 2 Boulevard Lavoisier, 49045 Angers Cedex 01, France

[2]Department of Ocean Systems, NIOZ Royal Netherlands Institute for Sea Research and Utrecht University, Den Burg, The Netherlands

[3]Department of Estuarine & Delta Systems, NIOZ Royal Netherlands Institute for Sea Research and Utrecht University, Korringaweg 7, 4401 NT Yerseke, The Netherlands.

[4]Department of Analytical, Environmental, and Geo-chemistry, Vrije Universiteit Brussel, Pleinlaan 2, 1050 Brussels, Belgium

[5]Department of Earth Sciences (Geochemistry), Faculty of Geosciences, Utrecht University, 3508 TA Utrecht, The Netherlands

*Correspondence to*: Jassin Petersen (jassin.petersen@univ-nantes.fr)

**Abstract.** The adaptation of some benthic foraminiferal species to low oxygen conditions provides the prospect of using the chemical composition of their tests as proxies for bottom water oxygenation. Manganese may be particularly suitable as such a geochemical proxy, because this redox element is soluble in reduced form ($Mn^{2+}$), and hence can be incorporated into benthic foraminiferal tests under low oxygen conditions. Therefore, intra- and inter-test differences in foraminiferal Mn/Ca ratios may hold important information about short term variability in pore water $Mn^{2+}$ concentrations and sediment redox conditions. Here, we studied Mn/Ca inter- and intra-test variability of living individuals of the shallow infaunal foraminifer *Ammonia tepida* sampled in Lake Grevelingen (The Netherlands) in three different months of 2012. The deeper parts of this lake are characterised by seasonal hypoxia/anoxia with associated shifts in microbial activity and sediment geochemistry, leading to seasonal $Mn^{2+}$ accumulation in the pore water. Earlier laboratory experiments with similar seawater $Mn^{2+}$ concentrations as encountered in the pore waters of Lake Grevelingen suggest that intrinsic intra-test variability in *A. tepida* (11-25 % RSD) is responsible for a considerable portion of the observed variability in Mn/Ca. Our results show that the seasonally highly dynamic environmental conditions in the study area lead to a strongly increased Mn/Ca intra- and inter-test variability (average of 45 % RSD). Within single specimens, both increasing and decreasing trends in Mn/Ca ratios with size are observed. Our results suggest that the variability of successive single chamber Mn/Ca ratios reflects the temporal variability of pore water $Mn^{2+}$. Additionally, active or passive migration of the foraminifera in the surface sediment may explain part of the observed Mn/Ca variability.

## 1 Introduction

In many coastal ecosystems, high summer temperatures and eutrophication lead to seasonally occurring hypoxia ($[O_2] < 63$ µM, Rabalais et al., 2002; Diaz & Rosenberg, 2008), linked to the emergence of water column stratification in combination with lower oxygen solubility and higher respiration rates in warmer waters (e.g., Keeling et al., 2010). On the seafloor,

oxygen is consumed by respiration of marine biota, thereby relating bottom water oxygenation (BWO), benthic ecosystem functioning and organic carbon cycling (Altabet et al., 1995; Levin et al., 2009; Koho et al., 2013).

Most of the organic matter (OM) deposited onto the seafloor is mineralised in surface sediments by respiration processes, involving oxygen, nitrate, manganese and iron oxides or sulphate as electron acceptors (Froelich et al., 1979). If Fe and Mn (hydr)oxides are sufficiently abundant, their reduction can be relevant for the total OM decomposition in the sediment (Aller,

1990; Burdige, 1993; Canfield, 1993; Vandieken et al., 2006). When the bottom waters of coastal water bodies are oxygenated, Mn oxides are present in the oxic surface layer of the sediment, and are reduced to soluble $Mn^{2+}$ in the deeper anoxic sediment layers. Dissolved $Mn^{2+}$ can then diffuse upwards in the pore water across the oxic–anoxic boundary, where it precipitates again in the form of Mn oxides, leading to a continuous cycling of Mn within the upper sediment (Aller, 1994; Slomp et al., 1997). The sedimentation rate and the bioturbation intensity are important factors controlling the pore water

cycling of oxidized metal oxides in sediments (van de Velde and Meysman, 2016). Macrofaunal bioturbation may introduce Mn oxides in the deeper, anoxic sediment, where these minerals are subsequently reduced (Mouret et al., 2009; Thibault de Chanvalon et al., 2016). When eutrophication and stratification of the water column lead to hypoxic BWO conditions, the oxygen penetration depth (OPD) is reduced, thus diminishing the chance that pore water $Mn^{2+}$ is oxidized. In extreme cases, $Mn^{2+}$ may diffuse from the pore water into the water column (Sundby and Silverberg, 1985; Thamdrup et al., 1994; Dellwig

et al., 2007; Konovalov et al., 2007; Pakhomova et al., 2007; Kowalski et al., 2012).



Sedimentary records of manganese may reflect a variety of past environmental conditions, including bottom water redox state, continental runoff, surface water productivity and bottom water current dynamics (Reichart et al., 1997; Van der Weijden et al., 2006; Lenz et al., 2015). The use of sedimentary manganese as a proxy is complicated because of its complex biogeochemical dynamics, including remobilisation of once precipitated manganese oxides, and subsequent diagenetic

overprinting (Schenau et al., 2002). Mn incorporated in the calcite of benthic foraminifera potentially represents a more reliable proxy of redox conditions, since these marine protists build their shells (tests) in the upper sediment layer, where the presence of $Mn^{2+}$ may be a function of bottom water oxygenation, and $Mn^{2+}$ can be permanently incorporated into the tests. However, the ecology of the various foraminiferal species is crucial and can influence manganese incorporation in the shell, since different adaptation strategies to changes in the availability of OM and of oxygen lead to different microhabitats and

probably also to different calcification depths and periods (Jorissen et al., 1995; Van der Zwaan et al., 1999; Koho et al., 2015). So far, few studies have investigated the potential of using Mn/Ca ratios in benthic foraminifera as a proxy for bottom water redox state. These studies do show that benthic foraminifera adequately register environmental $Mn^{2+}$ concentrations in their tests (Reichart et al., 2003; Munsel et al., 2010; Glock et al., 2012; Groeneveld & Filipsson, 2013; Koho et al., 2015, 2017; McKay et al., 2015; Barras et al., subm.). For example, Koho et al. (2015; 2017) demonstrated that foraminiferal

species occupying a relatively deep microhabitat display higher Mn/Ca ratios than those living near the sediment surface. Furthermore, when using microanalytical techniques, capable of measuring elemental concentrations in single foraminiferal chambers, short term variability in oxygenation may be unraveled. Recent studies have shown the potential of intra-test variability in Mn/Ca to resolve vertical migration in the sediment and/or seasonal changes in oxygenation (Glock et al., 2012; McKay et al., 2015).

In order to apply Mn/Ca ratios in benthic foraminiferal tests as a quantifiable proxy of palaeo-redox conditions, it is necessary to assess the presence of ontogenetic trends (i.e., size-related effects) and/or other vital effects occurring during calcification; the variability due to such biological processes will hereafter be termed "intrinsic (intra-test) variability". This



intrinsic part of the total variability determines the threshold value above which the variability within a single specimen can

be recognised successfully as a response to environmental conditions. In case the total intra-test variability is greater than the

intrinsic variability, this larger part can be ascribed to environmental factors, and/or active or passive migration of the

foraminifera in the sediment surface layer. In such cases, single chamber measurements could provide information about the

temporal variability of the Mn dynamics. In fact, a culturing study of *Ammonia tepida* and *Bulimina marginata* yielded the

first results of intrinsic variability of Mn/Ca ratios for different concentrations of dissolved $Mn^{2+}$ in seawater (Barras et al.,

subm.). This culturing study found that Mn/Ca intrinsic variability varied between 11-25 % for *A. tepida* and a range of

seawater $Mn^{2+}$ concentrations similar to those found in the pore water of surface sediments in our study area.

Here, we investigate Mn/Ca intra- and inter-test variability in the same benthic foraminifer *A. tepida* as studied by Barras et

al. (subm.) in culturing experiments from field samples of Lake Grevelingen (The Netherlands). The shallow infaunal *A.

tepida* represents a species complex including several pseudocryptic species (Hayward et al., 2004; Schweizer et al., 2011;

Saad and Wade, 2016). In Lake Grevelingen, *A. tepida* is almost exclusively represented by the T6 genotype. *Ammonia

tepida* is abundant in coastal areas of temperate climate zones, tolerating diverse biological and environmental stress factors,

including low oxygen conditions (Moodley and Hess, 1992; Sen Gupta et al., 1996; Geslin et al., 2014; Nardelli et al., 2014;

Thibault de Chanvalon et al., 2015; Cesbron et al., 2016). The salty bottom waters of Lake Grevelingen, an artificial lake

created after the closure of a branch of the Rhine-Meuse-Scheldt estuary, are characterised by seasonal hypoxia ($[O_2] < 63$

$\mu M$) and anoxia ($[O_2] <$ detection limit of $1 \mu M$) (Hagens et al., 2015; Seitaj et al., 2017). Therefore, samples from this site

represent a suitable context to study short term environmental variability (on a time scale of weeks to months) in relation to

elemental incorporation into benthic foraminiferal tests. However, one complicating factor is that it has recently been shown

how the seasonal pattern of sediment geochemical cycles in Lake Grevelingen is strongly influenced by the activity of cable

bacteria. These are electrogenic bacteria that induce strong shifts in the sediment geochemistry (Seitaj et al., 2015, 2017,

Sulu-Gambari et al., 2016a, 2016b), and lead to substantial $Mn^{2+}$ mobilisation in pore waters (Rao et al., 2016; Sulu-Gambari

et al., 2016b; van de Velde et al., 2016). These cable bacteria tend to have a highly patchy distribution, which hence could

complicate the interpretation (Sulu-Gambari et al., 2016b; Seitaj et al., 2017). Our approach is to investigate the Mn/Ca

intra- and inter-test variability of *A. tepida*, by measuring individual chambers with laser ablation ICP-MS (LA-ICP-MS) for

selected living specimens sampled in three different months of 2012. We will compare the obtained Mn/Ca intra-test

variability with results on strictly intrinsic intra-test variability for the same species, measured for specimens from laboratory

experiments by Barras et al. (subm.), to find out whether the seasonal variability of BWO and pore water $Mn^{2+}$

concentrations has left an imprint on benthic foraminiferal Mn/Ca ratios.

## 2 Material and methods

### 2.1 Study area

Sediment samples were recovered from Lake Grevelingen at a single location (51°44.956 N, 03°53.826 E, water depth 23.1

m). The sampling site experiences seasonal hypoxia; for the year 2012, monthly recordings of water column oxygen

concentrations (Hagens et al., 2015), sedimentary microbial community composition (Seitaj et al., 2015), pore water

geochemistry (Sulu-Gambari et al., 2016a, 2016b) and benthic $O_2$ uptake rates (Seitaj et al., 2017) are available. In 2012,

BWO started to decrease in April and attained a minimum of about 20 µM (~8 % saturation) in August. After

homogenisation of the water column in September/October, BWO quickly rose to values of ~200 µM (~80 % saturation).

Pore water $Mn^{2+}$ showed highest concentrations (up to 310 µM) in winter and early spring followed by considerably lower

concentrations in summer and autumn (Sulu-Gambari et al., 2016b).

### 2.2 Samples of living benthic foraminifera

Living specimens of *Ammonia tepida* were sampled in Lake Grevelingen in March, July and September 2012. In all three

months, surface sediments of the sampling site were inhabited by dense populations of *A. tepida*. Living specimens were

recognized by CellTracker Green (CTG, Bernhard et al., 2006). CTG was applied on board R/V Luctor within one hour of

retrieval (Langlet, 2014). Adult specimens (size fraction 150-315 μm) from the sediment depth 0-0.5 cm were selected for LA-ICP-MS measurements. For March, July and September 10, 16 and 18 specimens were analysed, respectively. Prior to LA-ICP-MS analyses, all specimens were cleaned to remove sediment adherences (Barker et al., 2003) by rinsing them three times with ultra-pure water in 200 μL tubes, followed by one rinse in methanol and three final rinses in ultra-pure water.

During each rinse, the samples were gently agitated with a vortex machine.

**2.3 LA-ICP-MS operating conditions and instrument calibration**

For all specimens several consecutive test chambers were measured individually by LA-ICP-MS (Fig. 1). The analyses were performed with an ArF excimer laser (193 nm, Analyte G2, Photon machines Inc.) coupled to a quadrupole ICP-MS (820-MS, Varian) at the Laboratory of Planetology and Geodynamics, Université de Nantes (France). Ablations were conducted

in a HelEx 2 Volume Cell with He as a carrier gas, a laser energy density of 0.91 J/cm² and a repetition rate of 4 Hz. To maximise the amount of ablated material, spot sizes were adapted to the chamber size and varied typically between 40 and 85 μm in diameter. The LA-ICP-MS operating conditions are summarised in Table 1 and the isotope masses selected for analyses were $^{24}Mg$, $^{27}Al$, $^{43}Ca$, $^{55}Mn$, $^{57}Fe$, $^{66}Zn$, $^{88}Sr$, and $^{137}Ba$.

Prior to each analytical session, the ICP-MS was tuned with the NIST SRM 612 reference material to minimise oxide

formation ($ThO^+/Th^+ < 0.5$ %), and elemental fractionation (U/Th close to 1), as well as to optimise the signal to noise ratio for Mn. The typical laser ablation profile of a foraminiferal chamber includes 30 s of data acquisition of the background signal (laser turned off) followed by the ablation of the chamber wall until it was completely pierced (the laser was shut down after a visual control of the ablation), and data acquisition was stopped after another measuring interval of the background signal. The NIST SRM 612 glass reference material was analysed every 10 foraminiferal measurements and the

NIST SRM 610 (silicate glass reference material), USGS MACS-3 (carbonate reference material) and the NFHS (NIOZ, Netherlands Institute of Sea Research, foraminifera in-house standard, Mezger et al., 2016) were analysed every 20


analytical spots. All reference materials have been analysed in raster mode, with the same laser energy as for the samples, a spot size of 65 µm and a scan speed of 10 µm/s. Because data acquisition on reference materials is on average 3 times longer (30 s) compared to foraminifera data (~10 s), the choice of raster mode for reference materials allows to keep the depth of laser ablation holes in reference materials similar as for the spot ablation of the foraminiferal chamber walls, minimising

differences in elemental fractionation linked to crater depth (Eggins et al., 1998). All foraminiferal ablation profiles were normalised to $^{43}$Ca as internal standard, and element concentrations were calculated assuming 40 % wt for the CaCO$_3$. The NIST SRM 612 glass served as calibration standard for Mn/Ca and Sr/Ca of the foraminiferal samples, using the recommended values of Jochum et al. (2011). The results obtained from NIST SRM 610, USGS MACS-3 and NFHS were also normalised to the NIST SRM 612 glass to evaluate the long term reproducibility of our analyses (Table 2). For each

element and analytical session we have calculated the Limit of Detection (LOD = background signal + 3.3*σ standard deviation calculated on the background signal) and the Limit of Quantification (LOQ = background signal + 10*σ), and have discarded all data below the LOQ (Long & Winefordner, 1983; Longerich et al., 1996; R. Bettencourt da Silva & A. Williams, 2015). As a result of the low energy laser ablation conditions, $^{24}$Mg profiles of NIST SRM 612 were below the LOQ for some datasets. However, the USGS MACS-3 has high Mg concentrations (1880 ±70 ppm), resulting in signals

above the LOQ, and thus Mg/Ca ratios of foraminiferal samples, NFHS and NIST SRM 610 were normalised to USGS MACS-3 using the recommended values of Jochum et al. (2012).

**2.4 Data treatment**

All laser ablation profiles for each element and each sample or reference material were carefully examined and processed with GLITTER software. Firstly, the integration interval was based on constant raw counts of $^{44}$Ca as indicator of calcite

ablation and on constant Sr/Ca ratios, as the intra-shell heterogeneity of Sr/Ca in foraminifera in general and of *A. tepida* in specific is known to be relatively small (e.g., Eggins et al., 2003; de Nooijer et al., 2014a). Secondly, each laser ablation profile was screened for peaks in elements that may indicate surface contamination ($^{27}$Al, $^{57}$Fe, $^{66}$Zn). Typically, high

concentrations of Mn on the outer and inner shell surfaces are considered as an indicator of contamination (Marr et al., 2011; de Nooijer et al., 2014a; Leduc et al., 2014; Koho et al., 2015). In our case, where Mn is the element of interest, whenever peaks of Mn on outer and inner parts of profiles corresponded to peaks of other contaminant elements, they were discarded from our data. All integration intervals containing less than 10 data points or results with count rates below the LOQ were

removed from our dataset.

The external reproducibility of our Mn/Ca analyses was 2 % (calculated as 2*RSE, relative standard error) when determined on the NFHS foraminiferal carbonate standard (data for Mn/Ca, Mg/Ca and Sr/Ca in Table 2). This standard is adequate for this purpose since it has Mn concentrations comparable to our samples and the same matrix. The results for the USGS MACS-3 carbonate and NIST SRM 610 glass reference materials (Table 2) agree with the recommended values (published

values have RSDs, relative standard deviations, of 4 % and 1%, respectively, Table 2). Additionally, we analysed the GJR calcite standard 4 times and the calculated concentrations for Mn and Sr (107 ±1 ppm and 190 ±3 ppm, respectively) compared well with published values (106 ±7 ppm and 184 ±15 ppm, respectively) from Wit et al. (2010).

**2.5 Statistical analyses**

Statistical analyses were carried out using R (R Core Team, 2016) using the package ggplot2 for graphical representation

(Wickham, 2009). To verify if data were normally distributed we used the Shapiro-Wilk test. For normally distributed data we used ANOVA and t-tests with the Bonferroni adjustment as post-hoc test. In all other cases we used a Kruskal-Wallis test and Wilcoxon-Mann-Whitney as post-hoc test. When comparing Mn/Ca ratios between chambers, in order to check for ontogenetic trends, we used Spearman rank correlation. In all cases, a p-value below 0.05 was considered as significant.

## 3 Results

### 3.1 Average Mn/Ca ratios per specimen

For all specimens of *A. tepida* investigated, between four and nine chambers per specimen fulfilled our profile selection

criteria for reliable Mn/Ca measurements (as defined in section 2.4). Mn/Ca ratios of individual chambers ranged from 0.03

to 0.57 mmol/mol for the 10 specimens sampled in March 2012 (Fig. 2A, Table A.1), from 0.04 to 0.63 mmol/mol for the 16

specimens collected in July 2012 (Fig. 2B, Table A.1), and from 0.04 to 0.48 mmol/mol for the 18 specimens sampled in

September 2012 (Fig. 2C, Table A.1). Average values for Mn/Ca per specimen were between 0.08 ±0.02 and 0.28 ±0.15

mmol/mol in March, between 0.08 ±0.04 and 0.39 ±0.19 mmol/mol in July and between 0.09 ±0.03 and 0.22 ±0.13

mmol/mol in September (Fig. 3A-C, Table A.1). There are no significant differences in average Mn/Ca ratios per specimen

between the three sampled months (Fig. 3D; ANOVA, Table 3). However, for the standard deviations per specimen, the

values from July 2012 are significantly larger than those of the two other months (ANOVA, Table 3).

### 3.2 Intra-test variability

The intra-test variability defined as RSD per specimen is on average 45 % (±15 % SD for all specimens) for Mn/Ca, 49 %

(±24 %) for Mg/Ca and 9 % (±4 %) for Sr/Ca. (Table A.1, A.2 and A.3). The specimens with the highest Mn/Ca RSD (three

specimens with RSD > 70%) were sampled in July (specimens 15, 17 and 18, Fig. 2B). Out of these three specimens, two

had one chamber with much higher Mn/Ca compared to the other chambers (specimens 15 and 17), whereas one specimen

had an increase in Mn/Ca ratios from older to younger chambers (specimen 18).

To further investigate the variability of Mn/Ca ratios within single specimens, we calculated the range (maximal minus

minimal chamber-value) for each measured individual. The so defined Mn/Ca intra-test variability shows a different pattern

for all specimens from each of the three sampling campaigns (Fig. 4, data in Table A.1). For instance, in March seven out of

ten specimens had a range smaller than 0.2 mmol/mol, whereas the remaining three had a maximum Mn/Ca intra-test

variability up to 0.5 mmol/mol. Yet, for the specimens sampled in July, twelve out of sixteen had a range larger than 0.2 mmol/mol with one specimen showing a range of 0.5-0.6 mmol/mol. Most specimens collected in September had a Mn/Ca range within specimens of 0.1-0.2 mmol/mol, with no specimen exceeding a difference in Mn/Ca of 0.4 mmol/mol. Because the range histogram for March clearly shows a non normal distribution (p-value = 0.041) and the sample size per sampling

date is small (n between 10 and 18), we investigated the differences with a non-parametric test and obtained a significant difference between July and the other two months. This result confirms the higher Mn/Ca intra-test variability in July compared to March and September (Table 3).

**3.3 Ontogenetic trends of Mn/Ca ratios**

To investigate the presence of ontogenetic (i.e., size-related) trends, we distinguished between four types: (a) specimens with

a trend towards lower values in later chambers (n=7; ID 1, 2, 5, 6, 9, 14 and 31 in Fig. 2), (b) specimens that tended to have higher Mn/Ca ratios in later chambers (n=6; ID 18, 23, 28, 30, 34 and 39 in Fig. 2), (c) specimens with only one or two values deviating from generally rather constant Mn/Ca ratios in all other chambers (n=2; ID 17 and 19 in Fig. 2), and (d) specimens with no apparent trend (the remaining 29 specimens). Most specimens of the first group were sampled in March 2012; in fact half of the specimens from March showed a tendency towards lower Mn/Ca ratios in later chambers. Most

individuals with increasing values towards later chambers were collected in September 2012. In the sample of July 2012 no dominant trend was observed.

To check for the existence of persistent ontogenetic trends, we combined for each successive chamber the values of all measured specimens and tested for statistical significant trends (Fig. 5). As expected, for March 2012 there is a significant trend towards lower values in later chambers (Fig. 5A). In July 2012 no significant trend is found (Fig. 5B) whereas in

September 2012 there was a slight, significant trend of increasing values towards later chambers (Fig. 5C). If all specimens from the three sampled months are combined (Fig. 5D), there is no significant relation with chamber stage.



## 4 Discussion

### 4.1 Comparison of Mn/Ca ratios between benthic foraminiferal species from coastal and deep-sea ecosystems

Recently, Mn/Ca ratios in benthic foraminiferal tests have been proposed as a potential palaeo-proxy for BWO. This suggestion is based on several observations made in recent deep-sea ecosystems (Reichart et al., 2003; Glock et al., 2012;

Groeneveld and Filipsson, 2013; Koho et al., 2015, 2017). Furthermore, the application of foraminiferal Mn/Ca ratios as a proxy for dissolved Mn has been tested in laboratory conditions, where calibration studies show a linear relation between seawater dissolved $Mn^{2+}$ concentrations and foraminiferal Mn/Ca ratios, though with species-dependent partition coefficients (Munsel et al., 2010; Barras et al., subm.). The use of foraminiferal Mn/Ca ratios as a palaeo-proxy for BWO has further been explored in several studies of deep-sea sediment records (Klinkhammer et al., 2009; Ní Fhlaithearta et al., 2010;

McKay et al., 2015).

Our study is the first to investigate benthic foraminiferal Mn/Ca ratios in a coastal ecosystem. The results for *A. tepida* show an average Mn/Ca ratio of 0.17 ±0.08 mmol/mol and a range of 0.08 ±0.04 to 0.39 ±0.19 mmol/mol for the average Mn/Ca per specimen (Table A.1). This range is comparable to that found in living specimens of some deep-sea infaunal species from the NE Japan margin (*E. batialis, B. spissa, U. cf. graciliformis, U. akitaensis, N. labradorica*; 0.0020 to 0.277

mmol/mol; Koho et al., 2017), but is elevated compared to Mn/Ca ratios measured in single tests of living benthic foraminifera from the Peruvian OMZ (*B. spissa*; 0.0021 to 0.010 mmol/mol; Glock et al., 2012). This latter difference can be explained by the generally higher pore water $Mn^{2+}$ concentrations in our study (< 310 µmol/L) and in the study of Koho et al. (2017; <5 µmol/L) compared to the study of Glock et al. (2012; <0.1 µmol/L). Despite the different pore water $Mn^{2+}$ concentrations in our study compared to Koho et al. (2017), we found rather similar foraminiferal Mn/Ca ratios, and this

contrast could be partly resolved by higher partition coefficients in deep-sea species compared to coastal species (Barras et al., subm.). For *A. tepida*, the range measured in specimens from Lake Grevelingen in our study compares well with measured Mn/Ca ratios in the study of Barras et al. (subm.; 0.13 and 0.86 mmol/mol) for a similar range of pore water $Mn^{2+}$

concentrations compared to the concentration of dissolved Mn in the seawater of the culturing experiments (10 to 100 µmol/L).

## 4.2 Intra-test variability of elemental ratios in benthic foraminiferal tests

Intra-test variability is less well documented for Mn/Ca ratios than for other elemental ratios (el/Ca; e.g., Mg/Ca, Sr/Ca) and

it has not yet been established what portion of the total variability can be attributed to either intrinsic (i.e., intra-test variability due to ontogenetic trends or other vital effects) or environmental factors (e.g., seasonality of Mn cycling in the surface sediments, microhabitat effects). In fact, it is essential to know what degree of intrinsic variability can be expected in a population having experienced the same environmental conditions, and above what threshold changes in Mn/Ca ratios can be ascribed to environmental factors. When using LA-ICP-MS, or other microanalytical techniques, Mn/Ca ratios are

measured on small parts of the foraminiferal test, making knowledge of intra-test variability even more crucial for the interpretation of the measurements. In case of a high intrinsic variability, independent of environmental parameters, more spot measurements will be necessary to obtain a reliable mean value for one specimen or several specimens from the same stable environment (Sadekov et al., 2005; de Nooijer et al., 2014a). Although contamination is an important issue in Mn/Ca measurements (Boyle, 1983; Barker et al., 2003; Pena et al., 2005, 2008), pre-treatment cleaning and a precise targeting of

the measurement interval (when using secondary ion mass spectrometry, SIMS, or LA-ICP-MS), should largely eliminate the potential influence of contaminant phases and/or diagenetic overgrowths (Glock et al., 2012; Koho et al., 2015; McKay et al., 2015). Therefore, the Mn/Ca intra-test variability should ideally not have a diagenetic contribution.

The results for all specimens measured in this study showed an average Mn/Ca intra-test variability for *A. tepida* of 45 ±15 % (RSD average for all chambers measured on a single specimen, Table A.1), which is comparable to that reported in some

previous studies (30-50 %; Glock et al., 2012; McKay et al., 2015; Koho et al., 2017). Our results from the different sampling campaigns showed more specimens with higher variability in July compared to March and September (Fig. 2 and



Fig. 4). In order to investigate if this significant difference can be attributed to environmental factors, our results will be compared to variability reported in other elements.

### 4.2.1 Approach for estimating (intrinsic) intra-test variability of Mn/Ca, Mg/Ca and Sr/Ca

The average Mn/Ca intra-test variability observed for *A. tepida* (45 %) is comparable to that measured for Mg/Ca (49 %,

Table A.2) but is larger than for Sr/Ca (9 %, Table A.3). Similarly, in other field studies of benthic foraminifera, average intra-test variability was ~20-50 % RSD for Mg/Ca (Allison & Austin, 2003; Curry & Marchitto, 2008; Raitzsch et al., 2011), and 5 % RSD for Sr/Ca (Allison & Austin, 2003). In these studies it could not be determined how much of the observed variability was due to intrinsic and environmental factors, respectively. In contrast to field studies, the use of cultured foraminifera offers the advantage that specimens have grown under exactly the same stable physico-chemical

conditions (no environmental variability), thus intra-test variability of elemental ratios is entirely due to biological processes. These culture studies suggest that when only intrinsic intra-test variability is considered, the RSD is ~30 % for Mg/Ca, compared to ~8 % for Sr/Ca (values for all chambers analysed by LA-ICP-MS, de Nooijer et al., 2014a). This difference between intrinsic intra-test variability of Mg/Ca and Sr/Ca was explained by the fact that during biomineralisation processes, unlike Sr, Mg is strongly discriminated against in the calcifying fluid (Bentov and Erez, 2006; Nehrke et al., 2013; de

Nooijer et al., 2014b). In some field studies (Allison and Austin, 2003; Curry and Marchitto, 2008), values for Mg/Ca intra-test variability were similar to those observed in the culturing experiment, and only a single field study (Raitzsch et al., 2011) reported substantially higher Mg/Ca intra-test variability, with an RSD of 51%. In the case of our *A. tepida* specimens, the intra-test variability of Mg/Ca was similarly high (49 %). Contrastingly, the Sr/Ca RSD of *A. tepida* was 9 %, comparable to variability in Sr/Ca from culturing studies.



### 4.2.2 Intrinsic intra-test variability of Mn/Ca and ontogenetic trends

Similar as for Mg/Ca ratios it is possible to quantify the intrinsic intra-test variability of Mn/Ca based on results from cultured benthic foraminifera. Culture experiments performed with *A. tepida* in controlled and stable conditions show 85 % of variability (RSD) for very low $Mn^{2+}$ concentrations of 2 µmol/L (all chambers calcified under laboratory conditions,

measured by LA-ICP-MS, Barras et al., subm.). However, for seawater $Mn^{2+}$ concentrations of 12 and 101 µmol/L, comparable to the pore water concentration at our site, the variability (RSD) in Mn/Ca was 25 and 11 %, respectively (Barras et al., subm.). For even higher seawater $Mn^{2+}$ concentrations of 595 µmol/L the Mn/Ca RSD was 17 % (Barras et al., subm.). Despite analytical considerations, the large RSD observed for the lowest $Mn^{2+}$ concentration was partly due to the presence of a clear decrease of Mn/Ca during the ontogeny. At higher concentrations, no such ontogenetic tendencies were found

(Barras et al., subm.). In our material there were no systematic ontogenetic trends either (Fig. 2 and 5), so that the 11-25 % RSD range measured for $Mn^{2+}$ concentrations of ~10-100 µM, respectively, should be representative for the intrinsic intra-test variability of our specimens. This range is much lower than the average total intra-test variability of 45 % found for living *A. tepida* in our study of Lake Grevelingen.

Consequently, it appears that at most, about half of the total variability in Mn/Ca can be attributed to intrinsic factors. The

remaining part of the variability in Mn/Ca should be due to changing pore water $Mn^{2+}$ concentrations in the calcification environment of the foraminifera. This may be due to environmental changes during the lifespan of the individuals, or, alternatively, to active or passive vertical foraminiferal migration through different biogeochemical micro-environments.

### 4.2.3 Seasonality of environmental factors as explanation for Mn/Ca intra-test variability

As explained in the previous paragraph, it appears that half or more of the measured Mn/Ca intra-test variability of *A. tepida*

(total intra-test variability was 45 % RSD compared to 11-25 % RSD intrinsic variability) can be attributed to environmental parameters. Here, we will consider which specific factors may be responsible for this variability. First of all, the sampling



site of *A. tepida* in Lake Grevelingen at 23.1 m water depth shows strong seasonal fluctuations in bottom water oxygenation

(section 2.1, Hagens et al., 2015). In theory, under oxic conditions the zone of manganese reduction should lie below the

microhabitat of *A. tepida*, which is situated close to the sediment-water interface (SWI) (Thibault de Chanvalon et al., 2015;

Cesbron et al., 2016), and chambers calcified in this condition should show low Mn/Ca ratios. When oxygen conditions

change (for instance due to enhanced fluxes of organic matter to the seafloor, or stagnation of bottom waters), the zone of

manganese reduction will migrate upward; under hypoxic conditions the Mn redox front will be situated closer to the SWI

and more $Mn^{2+}$ should be incorporated into chambers of *A. tepida*. However, in extreme cases, the upward migration of the

Mn redox front can lead to high amounts of $Mn^{2+}$ diffusing into the water column (Sundby and Silverberg, 1985; Konovalov

et al., 2007; Pakhomova et al., 2007), and in case of seafloor anoxia, $Mn^{2+}$ may almost totally seep out of the sediment into

the bottom water (Slomp et al., 1997). Thus, foraminiferal Mn/Ca ratios from chambers calcified under anoxic conditions

should be nominal. At our station in Lake Grevelingen, which is experiencing hypoxic but no anoxic events in summer, we

would expect higher $Mn^{2+}$ concentrations below the SWI in summer than in winter coinciding with higher Mn/Ca ratios in

chambers of *A. tepida* calcified in summer. However, contrary to these theoretical expectations, observations showed

maximum pore water $Mn^{2+}$ concentrations in the topmost cm from February to April 2012 whereas $Mn^{2+}$ concentrations

remained relatively low during the rest of 2012 (Sulu-Gambari et al., 2016a, 2016b). Recently, similar seasonal $Mn^{2+}$ pore

water patterns at a nearby, but slightly deeper station (~500 m away, depth 34 m) in Lake Grevelingen were explained by the

presence of cable bacteria in winter and early spring, which are capable of performing electrogenic sulfur oxidation resulting

in the dissolution of FeS (Seitaj et al., 2015; Sulu-Gambari et al., 2016a, 2016b). Consequently, upward diffusing $Fe^{2+}$ is

oxidized by manganese oxides which produces an accumulation of $Mn^{2+}$ in the pore water in winter (Sulu-Gambari et al.,

2016a, 2016b). This process would be responsible for a temporal offset of ~4-6 months between minimum BWO and

maximum $Mn^{2+}$ concentrations. Although cable bacteria were not detected at our sampling station with fluorescence *in situ*

hybridization (FISH), which may be due to their patchy distribution (Sulu-Gambari et al., 2016b; Seitaj et al., 2017), the



similarity in pore water data between the two stations strongly suggests that, also at our sampling station, cable bacteria activity is responsible for the observed strong seasonality in pore water $Mn^{2+}$. Therefore, it seems probable that *A. tepida*, sampled in March, July and September 2012, respectively, were confronted with strongly different pore water $Mn^{2+}$ concentrations, and may also have experienced important fluctuations in $Mn^{2+}$ in their microenvironment during their

lifespan.

The foraminiferal life-span generally varies from 3 months to about 2 years (Boltovskoy and Lena, 1969; Murray, 1983). No precise data are available for *A. tepida*, although on the basis of field evidence, Goldstein & Moodley (1993) and Morvan et al. (2006) concluded that their longevity should be at least one year. However, during the lifespan, chamber formation is probably not a continuous process, and could be faster in juvenile stages (on the basis of lower metabolic rates in later

ontogenetic stages, inferred from increasing $\delta^{13}C$ ratios, e.g., Schmiedl et al., 2004; Mackensen, 2008; Schumacher et al., 2010; Raitzsch et al., 2011). Under experimental conditions, growth rates of *A. tepida* showed a decrease with ontogeny from 1 chamber per day to 1 chamber per week during a 3 weeks period (de Nooijer et al., 2014a). It has been suggested that at later ontogenetic stages, calcification could be limited to short periods with favorable conditions, such as abundant food supply (e.g., Jorissen, 1988). In view of the rather scarce information about the timing and duration of calcification, we

assume that *A. tepida* from Lake Grevelingen calcifies during several successive seasons, and that the specimens of our three samples (March, July and September 2012) have each recorded a different part of the yearly pore water $Mn^{2+}$ cycle. The different degrees of intra-test variability between sampled months (Fig. 3 and 4) could be explained by the fact that some specimens mainly calcify during periods with stable sediment $Mn^{2+}$ concentrations (i.e., a large part of specimens sampled in March and September), whereas others have calcified during periods with rapid changes, such as in winter, when cable

bacteria activity and $Mn^{2+}$ concentrations rapidly increase, or in spring, when these two parameters decrease again (i.e., a large part of specimens sampled in July). Different intra-test variability and different Mn/Ca trends (in successive chambers, Fig. 2 and 5) for different individuals of the same sample can then be explained by slightly different calcification histories.



### 4.2.4 Foraminiferal vertical migration as explanation for Mn/Ca intra-test variability

*Ammonia tepida* is a shallow infaunal taxon, preferring a microhabitat close to the SWI, implying that the chemical composition of its test in principle reflects conditions in the superficial sediment layer (Thibault de Chanvalon et al., 2015; Cesbron et al., 2016). Observations made in laboratory experiments show that specimens of this species introduced in deeper anoxic sediments rapidly decrease their activity (Langlet et al., 2013; Maire et al., 2016) and, according to a field study from the Loire estuary, only specimens from the top 2 cm appear to be capable to regain their preferred niche at the sediment surface (Thibault de Chanvalon et al., 2015). However, Geslin et al. (2014) and Barras et al. (subm.) have shown that several foraminiferal species (among which *A. tepida*) repeatedly calcify new chambers in hypoxic conditions, whereas Nardelli et al. (2014) showed that even in anoxic conditions, *A. tepida* is capable of producing at least one new chamber. These data suggest that individuals which have actively (through vertical migration) or passively (being transported by burrowing macrofauna) moved to deeper sediment layers, with higher $Mn^{2+}$ levels (such as in February and March 2012, when the OPD was ~0.2-0.3 cm below the SWI, Seitaj et al., 2016) could incidentally calcify a single chamber with a much higher Mn/Ca ratio. This could be the explanation for the patterns observed, for instance, in specimens 15, 17 and 19 sampled in July 2012 (Fig. 2). This phenomenon could be responsible for a small contribution to increased Mn/Ca intra-test variability (specimens 15 and 17 have high RSD values, Table A.1). In conclusion, we propose that a large part of the overall Mn/Ca intra-test variability can be explained by a different timing of calcification events with respect to the seasonal cycle of pore water $Mn^{2+}$ concentrations, whereas extremely high values in single chambers could be due to occasional chamber formation in deeper sediment layers.

### 4.3 Interest of benthic foraminiferal Mn/Ca ratios in coastal environments and implications for palaeoceanographic studies

Concerning the reliability of single chamber measurements, in view of the large intra-test variability, in the very dynamic environment of Lake Grevelingen it is necessary to measure several chambers to obtain reliable average Mn/Ca ratios for a



single specimen. In this specific context, bulk measurements could be a more practical solution to study the long term

evolution of BWO. However, the large intra-test variability may include very useful information about the seasonal

variability of pore water dynamics driven by redox conditions and sedimentation rates but also microbial activity. Such

information can only be obtained by spot measurements of individual chambers. In fact, a comparison of Mn/Ca ratios of

successive individual chambers can potentially inform us about the extent of these seasonal changes, and implicitly, about

cable bacteria activity. If the calcification season of the foraminifera could be determined more precisely (for instance by

using single specimen stable isotope analyses, e.g., Diz et al., 2009), it would potentially even be possible to reconstruct the

annual cycle of pore water dynamics.

**5 Conclusion**

Tests of the coastal benthic foraminifer *A. tepida*, sampled in three different months at 23.1 m depth in the seasonally

hypoxic/anoxic Lake Grevelingen show Mn/Ca ratios with a range of 0.08 ±0.04 to 0.39 ±0.19 mmol/mol (for average

values per specimen), associated with a very large intra-test variability (average RSD = 45 %). This high intra-test variability

may partly represent intrinsic factors (due to biological processes), although no systematic ontogenetic trends could be

identified. However, we ascribed the larger part of the Mn/Ca intra-test variability to the large temporal variability of

environmental parameters and to different timing of calcification of the analysed specimens. We suggest that the strong

seasonal dynamics of pore water Mn induced by seasonal hypoxia and cable bacteria activity, leading to variations in

absolute $Mn^{2+}$ concentrations and/or migration of the redox front in the sediment, is the main factor responsible for this large

intra-test variability. Differences in the timing of calcification could explain the different degrees of intra-test variability

observed for the three sampled months, whereas differences in individual life history (between the individuals found in a

single sample) could even explain inter-test differences observed in each of our three samples. Some individual chambers

with exceptionally high Mn/Ca ratios could be due to active or passive migration to, and calcification in, slightly deeper

sediment layers. In conclusion, in environments with strong seasonal changes in redox conditions and microbial activity,



Mn/Ca measurements of successive chambers of individual tests may provide a powerful proxy to reconstruct the seasonal

variability of these parameters.

**6 Acknowledgements**

This research was financially supported by the European Research Council under the European Union's Seventh Framework

Programme (FP/2007-2013) through ERC Grant 306933 to FJRM. The Region Pays de la Loire is thanked for financing the

MADONA project, including the PhD allocation of JP. We are very grateful to the SCIAM laboratory (University of Angers)

for the foraminiferal SEM image.





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





**Table 1: Summary of LA-ICP-MS operating conditions.**

| Analyte G2 laser ablation system (193 nm) | |
|---|---|
| Laser fluence | 0.91 J/cm$^2$ |
| Laser pulse repetition rate | 4 Hz |
| He flow rates for the HelEx 2 Volume cell | 0.7 and 0.3 L/min |
| Projected spot size | 40-85 µm |

| Quadrupole Varian 820-MS | |
|---|---|
| Dwell time | 20 ms |
| RF power | 1.15 kW |
| Sheath gas flow | 0.8-0.9 L/min |
| $^{232}Th^{16}O^+/^{232}Th^+$ | < 0.5% |
| U/Th | ~1 |

**Table 2: Evaluation of external reproducibility of laser ablation ICP-MS analyses for Mg/Ca, Mn/Ca and Sr/Ca determined on NFHS in-house foraminiferal standard, USGS MACS-3 and NIST SRM 610. For Mn/Ca and Sr/Ca all values are calibrated**
5 **against NIST SRM 612 and are given as average with standard deviations from all sessions (in bold). For Mg/Ca all values are calibrated against USGS MACS-3. Reference values with standard deviation are given according to Jochum et al. (2011), Jochum et al. (2012), USGS (S. Wilson, USGS, unpubl.) and Pearce et al. (1997). Values for NFHS are in mmol/mol for better comparison with results from samples (standard deviation of standard applied as error bar on samples) as well as in ppm because reference values are given in this unit (NIOZ, personal communication) and for comparison to concentrations of other reference materials.**
10 **Values for USGS MACS-3 and NIST SRM 610 are in ppm for comparison with published reference values.**

| | NFHS | | | | USGS MACS-3 | | | NIST SRM 610 | | |
|---|---|---|---|---|---|---|---|---|---|---|
| | this study | | this study | NIOZ (p. com.) | this study | USGS | Jochum et al. (2012) | this study | Jochum et al. (2011) | Pearce et al. (1997) |
| | [mmol/mol] | | [ppm] | [ppm] | [ppm] | [ppm] | [ppm] | [ppm] | [ppm] | [ppm] |
| Mg/Ca | 2.8 ±0.2 (n=44) | Mg | 688 ±40 | 660 | - | 1756 ±136 | 1880 ±70 (n=36) | 520 ±12 (n=19) | 432 ±29 | 465 ±27 |
| Mn/Ca | 0.15 ±0.02 (n=115) | Mn | 85 ±9 | 88 | 520 ±26 (n=115) | 536 ±28 | 532 ±23 (n=36) | 445 ±5 (n=35) | 444 ±6 | 433 ±32 |
| Sr/Ca | 1.38 ±0.04 (n=115) | Sr | 1204 ±37 | 1300 | 6590 ±383 (n=115) | 6760 ±350 | 6570 ±170 (n=36) | 513 ±16 (n=35) | 515.5 ±0.5 | 497 ±18 |





**Table 3: Statistical results for comparison of Mn/Ca measurements of *A. tepida* for different sampling months. Significant results (p-value < 0.05) are indicated in bold.**

|  | Average Mn/Ca | SD Mn/Ca | Range Mn/Ca |
|---|---|---|---|
| **p-value ANOVA** | 0.118 | **0.016** | - |
| **p-value Kruskal-Wallis** | - | - | **0.044** |
| **p-value post-hoc test:** |  |  |  |
| **03/2012 vs 07/2012** | 0.720 | **0.045** | **0.041** |
| **03/2012 vs 09/2012** | 1.000 | 1.000 | 0.621 |
| **07/2012 vs 09/2012** | 0.120 | **0.036** | **0.033** |



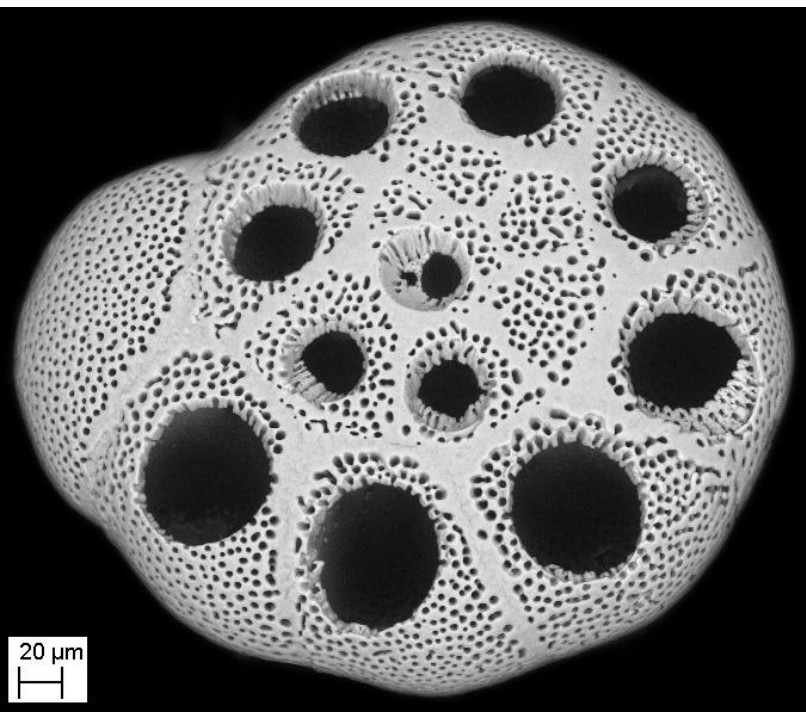

**Figure 1: SEM image of *A. tepida* specimen after laser ablation analysis of multiple chambers. Image taken at SCIAM, University of Angers.**





**Figure 2: Mn/Ca for each chamber in living specimens of *A. tepida* from Lake Grevelingen, collected in March (A), July (B) and September (C) 2012 (specimens labelled from 1 to 44 as in Table A.1). One plot represents one specimen. Numbers on x-axis indicate chambers: 1 = penultimate chamber 2 = antepenultimate chamber, etc., 10= central part of test. Error bars represent the Mn/Ca standard deviation of multiple analyses of a foraminiferal carbonate standard NFHS (see section 2.3).**





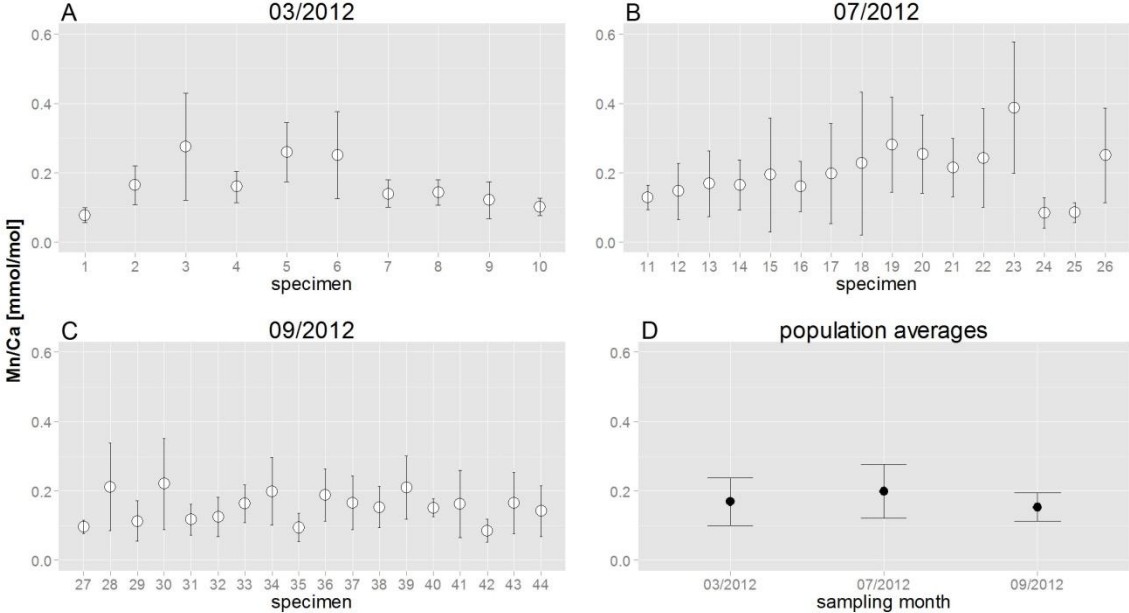

**Figure 3: Average Mn/Ca values and standard deviation for all measured chambers of each analysed specimen of *A. tepida* for the three sampling campaigns (A: March, B: July, C: September). D: Population averages with standard deviation calculated on the basis of mean values per specimen of all analysed specimens for each of the three sampling months.**





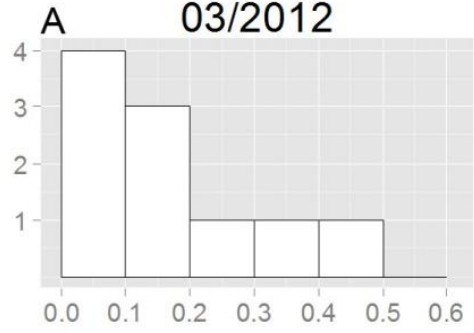

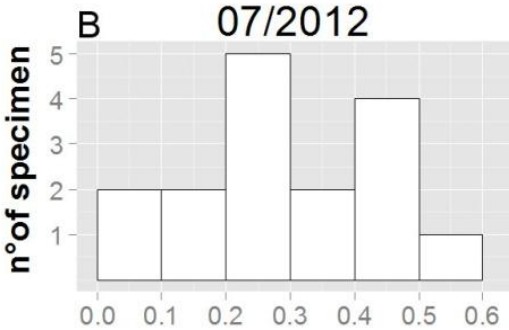

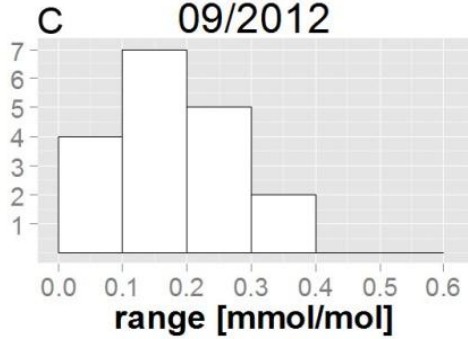

**Figure 4: Histograms representing Mn/Ca intra-test variability as range (difference between maximum and minimum Mn/Ca per specimen of *A. tepida*). A: March, 10 specimens; B: July, 16 specimens; C: September, 18 specimens.**



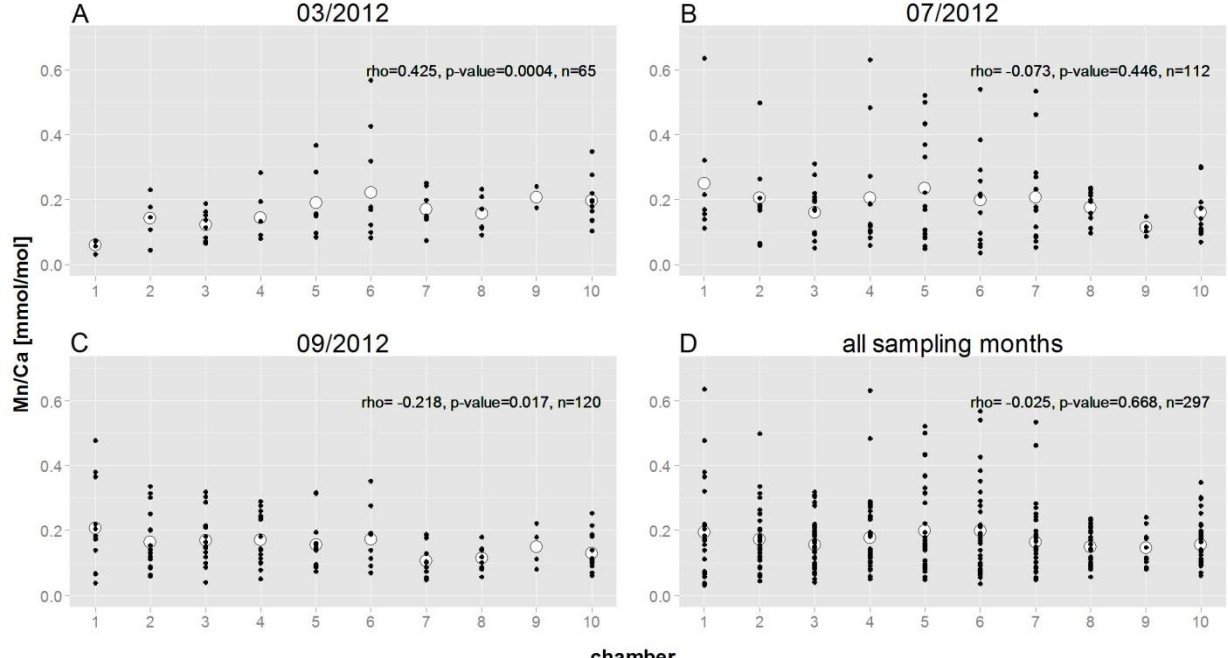

**Figure 5. Mn/Ca of all measured specimens as a function of chamber succession. A: March, B: July, C: September, D: all combined. Numbers on x-axis indicate chambers: 1 = penultimate chamber 2 = antepenultimate chamber, etc., 10= central part of test. Small black dots represent Mn/Ca of single measurements; larger white circles represent average values per chamber, when all specimens are combined. Trends were tested on statistical significance with Spearman rank correlation (correlation coefficient rho, p-value and n (total number of data points) are indicated).**



**Table A.1. Results of Mn/Ca measurements for different samplings of living specimens of *A. tepida* from Lake Grevelingen, station ST2. All data are normalised to SRM NIST 612. Values are calculated per specimen (ID corresponds to numbers attributed to specimens in Fig. 2). The number of chambers included is indicated in column "n".**

| sampling date | ID | n | Mn/Ca average [mmol/mol] | SD [mmol/mol] | RSD [%] | Mn/Ca max [mmol/mol] | min [mmol/mol] | range [mmol/mol] |
|---|---|---|---|---|---|---|---|---|
| 03/2012 | 1 | 8 | 0.08 | 0.02 | 27.4 | 0.10 | 0.03 | 0.07 |
| 03/2012 | 2 | 9 | 0.16 | 0.06 | 34.4 | 0.24 | 0.04 | 0.20 |
| 03/2012 | 3 | 7 | 0.28 | 0.15 | 56.3 | 0.57 | 0.11 | 0.45 |
| 03/2012 | 4 | 10 | 0.16 | 0.04 | 28.4 | 0.23 | 0.07 | 0.16 |
| 03/2012 | 5 | 5 | 0.26 | 0.09 | 32.9 | 0.35 | 0.15 | 0.20 |
| 03/2012 | 6 | 5 | 0.25 | 0.13 | 50.3 | 0.43 | 0.07 | 0.35 |
| 03/2012 | 7 | 4 | 0.14 | 0.04 | 28.3 | 0.17 | 0.08 | 0.09 |
| 03/2012 | 8 | 6 | 0.14 | 0.04 | 26.2 | 0.18 | 0.10 | 0.08 |
| 03/2012 | 9 | 5 | 0.12 | 0.05 | 44.3 | 0.20 | 0.06 | 0.14 |
| 03/2012 | 10 | 6 | 0.10 | 0.02 | 23.6 | 0.14 | 0.07 | 0.06 |
| 07/2012 | 11 | 8 | 0.13 | 0.03 | 27.1 | 0.17 | 0.08 | 0.08 |
| 07/2012 | 12 | 8 | 0.15 | 0.08 | 56.1 | 0.30 | 0.06 | 0.23 |
| 07/2012 | 13 | 8 | 0.17 | 0.09 | 56.1 | 0.32 | 0.06 | 0.26 |
| 07/2012 | 14 | 5 | 0.16 | 0.07 | 44.2 | 0.23 | 0.07 | 0.16 |
| 07/2012 | 15 | 7 | 0.19 | 0.16 | 85.0 | 0.53 | 0.06 | 0.48 |
| 07/2012 | 16 | 6 | 0.16 | 0.07 | 45.1 | 0.27 | 0.07 | 0.20 |
| 07/2012 | 17 | 9 | 0.20 | 0.14 | 72.7 | 0.54 | 0.08 | 0.46 |
| 07/2012 | 18 | 9 | 0.23 | 0.21 | 90.8 | 0.63 | 0.05 | 0.58 |
| 07/2012 | 19 | 7 | 0.28 | 0.14 | 49.0 | 0.50 | 0.19 | 0.31 |
| 07/2012 | 20 | 5 | 0.25 | 0.11 | 44.3 | 0.44 | 0.16 | 0.28 |
| 07/2012 | 21 | 8 | 0.21 | 0.08 | 39.2 | 0.33 | 0.10 | 0.24 |
| 07/2012 | 22 | 8 | 0.24 | 0.14 | 58.8 | 0.48 | 0.10 | 0.39 |
| 07/2012 | 23 | 4 | 0.39 | 0.19 | 49.1 | 0.63 | 0.17 | 0.46 |
| 07/2012 | 24 | 8 | 0.08 | 0.04 | 51.6 | 0.17 | 0.05 | 0.12 |
| 07/2012 | 25 | 5 | 0.09 | 0.03 | 33.9 | 0.11 | 0.04 | 0.07 |
| 07/2012 | 26 | 7 | 0.25 | 0.14 | 54.7 | 0.52 | 0.10 | 0.42 |
| 09/2012 | 27 | 5 | 0.10 | 0.02 | 19.3 | 0.12 | 0.07 | 0.05 |
| 09/2012 | 28 | 7 | 0.21 | 0.13 | 59.9 | 0.48 | 0.10 | 0.38 |
| 09/2012 | 29 | 9 | 0.11 | 0.06 | 51.2 | 0.23 | 0.05 | 0.18 |
| 09/2012 | 30 | 8 | 0.22 | 0.13 | 59.5 | 0.38 | 0.05 | 0.33 |
| 09/2012 | 31 | 6 | 0.12 | 0.04 | 37.9 | 0.18 | 0.06 | 0.12 |
| 09/2012 | 32 | 5 | 0.13 | 0.06 | 45.7 | 0.22 | 0.07 | 0.14 |
| 09/2012 | 33 | 4 | 0.16 | 0.05 | 33.3 | 0.24 | 0.11 | 0.13 |
| 09/2012 | 34 | 9 | 0.20 | 0.10 | 48.9 | 0.37 | 0.10 | 0.26 |
| 09/2012 | 35 | 8 | 0.09 | 0.04 | 43.4 | 0.14 | 0.04 | 0.10 |
| 09/2012 | 36 | 7 | 0.19 | 0.08 | 40.2 | 0.32 | 0.09 | 0.22 |
| 09/2012 | 37 | 5 | 0.17 | 0.08 | 46.3 | 0.24 | 0.07 | 0.18 |
| 09/2012 | 38 | 6 | 0.15 | 0.06 | 38.8 | 0.22 | 0.07 | 0.15 |
| 09/2012 | 39 | 6 | 0.21 | 0.09 | 43.7 | 0.30 | 0.10 | 0.20 |
| 09/2012 | 40 | 7 | 0.15 | 0.03 | 16.7 | 0.20 | 0.13 | 0.08 |
| 09/2012 | 41 | 7 | 0.16 | 0.10 | 60.3 | 0.32 | 0.04 | 0.28 |
| 09/2012 | 42 | 4 | 0.09 | 0.03 | 38.4 | 0.13 | 0.06 | 0.07 |
| 09/2012 | 43 | 8 | 0.17 | 0.09 | 53.4 | 0.30 | 0.07 | 0.24 |
| 09/2012 | 44 | 9 | 0.14 | 0.07 | 51.9 | 0.31 | 0.08 | 0.24 |
| | **average** | **6.8** | **0.17** | **0.08** | **45 (±15)** | **0.31** | **0.08** | **0.22** |



**Table A.2. Results of Mg/Ca measurements for different samplings of living specimens of *A. tepida* from Lake Grevelingen, station ST2. All data are normalised to USGS MACS-3. Values are calculated per specimen (ID corresponds to numbers attributed to specimens in Fig. 2). The number of chambers included is indicated in column "n".**

| sampling date | ID | n | Mg/Ca average [mmol/mol] | SD [mmol/mol] | RSD [%] | Mg/Ca max [mmol/mol] | min [mmol/mol] | range [mmol/mol] |
|---|---|---|---|---|---|---|---|---|
| 03/2012 | 1 | 8 | 1.62 | 0.20 | 12.2 | 2.06 | 1.43 | 0.63 |
| 03/2012 | 2 | 9 | 2.51 | 0.74 | 29.5 | 3.30 | 0.87 | 2.43 |
| 03/2012 | 3 | 7 | 4.16 | 2.48 | 59.6 | 8.21 | 1.67 | 6.54 |
| 03/2012 | 4 | 10 | 2.39 | 0.90 | 37.7 | 3.96 | 1.35 | 2.62 |
| 03/2012 | 5 | 5 | 3.78 | 0.89 | 23.5 | 4.52 | 2.55 | 1.97 |
| 03/2012 | 6 | 5 | 2.77 | 1.62 | 58.6 | 5.14 | 1.40 | 3.74 |
| 03/2012 | 7 | 4 | 2.47 | 1.42 | 57.4 | 4.56 | 1.41 | 3.16 |
| 03/2012 | 8 | 6 | 1.31 | 0.35 | 27.0 | 1.86 | 0.89 | 0.97 |
| 03/2012 | 9 | 5 | 1.83 | 0.66 | 36.1 | 2.52 | 0.78 | 1.73 |
| 03/2012 | 10 | 6 | 1.26 | 0.35 | 27.9 | 1.92 | 0.94 | 0.99 |
| 07/2012 | 11 | 8 | 1.19 | 0.42 | 35.5 | 1.97 | 0.82 | 1.15 |
| 07/2012 | 12 | 8 | 2.46 | 1.29 | 52.5 | 5.17 | 1.22 | 3.95 |
| 07/2012 | 13 | 8 | 1.99 | 1.55 | 78.3 | 4.73 | 0.73 | 3.99 |
| 07/2012 | 14 | 5 | 1.77 | 0.85 | 48.2 | 3.19 | 0.91 | 2.28 |
| 07/2012 | 15 | 7 | 3.12 | 2.75 | 88.3 | 7.21 | 0.74 | 6.47 |
| 07/2012 | 16 | 6 | 3.81 | 1.42 | 37.2 | 5.18 | 1.59 | 3.59 |
| 07/2012 | 17 | 9 | 4.92 | 1.29 | 26.2 | 6.44 | 2.55 | 3.89 |
| 07/2012 | 18 | 9 | 3.11 | 1.78 | 57.2 | 6.24 | 1.45 | 4.79 |
| 07/2012 | 19 | 7 | 5.69 | 2.32 | 40.7 | 9.47 | 2.40 | 7.07 |
| 07/2012 | 20 | 5 | 6.81 | 5.05 | 74.2 | 14.93 | 1.24 | 13.69 |
| 07/2012 | 21 | 8 | 3.34 | 2.24 | 67.0 | 7.36 | 1.12 | 6.23 |
| 07/2012 | 22 | 8 | 4.03 | 2.20 | 54.6 | 7.57 | 1.63 | 5.94 |
| 07/2012 | 23 | 4 | 7.71 | 3.58 | 46.4 | 10.42 | 2.71 | 7.71 |
| 07/2012 | 24 | 8 | 1.56 | 0.42 | 26.8 | 2.21 | 1.11 | 1.10 |
| 07/2012 | 25 | 5 | 1.71 | 1.13 | 65.9 | 2.95 | 0.73 | 2.22 |
| 07/2012 | 26 | 7 | 5.34 | 2.94 | 55.1 | 10.17 | 1.50 | 8.67 |
| 09/2012 | 27 | 5 | 2.24 | 1.74 | 77.6 | 5.32 | 1.09 | 4.23 |
| 09/2012 | 28 | 7 | 4.28 | 5.33 | 124.4 | 16.05 | 0.89 | 15.16 |
| 09/2012 | 29 | 9 | 1.84 | 0.60 | 32.7 | 2.51 | 0.93 | 1.58 |
| 09/2012 | 30 | 8 | 1.26 | 0.33 | 25.7 | 2.00 | 0.97 | 1.03 |
| 09/2012 | 31 | 6 | 1.64 | 0.31 | 19.1 | 1.99 | 1.11 | 0.88 |
| 09/2012 | 32 | 5 | 1.80 | 0.65 | 36.2 | 2.90 | 1.17 | 1.72 |
| 09/2012 | 33 | 4 | 2.29 | 0.14 | 6.3 | 2.45 | 2.14 | 0.30 |
| 09/2012 | 34 | 9 | 1.64 | 0.62 | 37.7 | 3.05 | 1.04 | 2.01 |
| 09/2012 | 35 | 8 | 1.73 | 0.69 | 40.0 | 3.10 | 1.12 | 1.98 |
| 09/2012 | 36 | 7 | 1.58 | 1.14 | 72.1 | 4.13 | 0.98 | 3.15 |
| 09/2012 | 37 | 5 | 1.29 | 0.59 | 45.8 | 2.08 | 0.72 | 1.36 |
| 09/2012 | 38 | 6 | 1.68 | 0.65 | 38.9 | 2.49 | 0.68 | 1.80 |
| 09/2012 | 39 | 6 | 3.46 | 1.53 | 44.2 | 5.58 | 1.40 | 4.18 |
| 09/2012 | 40 | 7 | 2.18 | 1.33 | 61.1 | 5.07 | 1.17 | 3.90 |
| 09/2012 | 41 | 7 | 4.32 | 4.71 | 109.0 | 13.88 | 0.91 | 12.98 |
| 09/2012 | 42 | 4 | 1.17 | 0.23 | 19.5 | 1.44 | 0.96 | 0.47 |
| 09/2012 | 43 | 8 | 2.69 | 1.86 | 68.8 | 7.08 | 1.33 | 5.75 |
| 09/2012 | 44 | 9 | 2.45 | 1.25 | 51.0 | 4.58 | 1.16 | 3.41 |
| **average** | | **6.8** | **2.78** | **1.47** | **49 (±24)** | **5.20** | **1.26** | **3.94** |



**Table A.3. Results of Sr/Ca measurements for different samplings of living specimens of *A. tepida* from Lake Grevelingen, station ST2. All data are normalised to SRM NIST 612. Values are calculated per specimen (ID corresponds to numbers attributed to specimens in Fig. 2). The number of chambers included is indicated in column "n".**

| sampling date | ID | n | Sr/Ca average [mmol/mol] | SD [mmol/mol] | RSD [%] | Sr/Ca max [mmol/mol] | min [mmol/mol] | range [mmol/mol] |
|---|---|---|---|---|---|---|---|---|
| 03/2012 | 1 | 8 | 1.36 | 0.17 | 12.6 | 1.59 | 1.00 | 0.59 |
| 03/2012 | 2 | 9 | 1.45 | 0.07 | 4.6 | 1.51 | 1.35 | 0.16 |
| 03/2012 | 3 | 7 | 1.43 | 0.08 | 5.5 | 1.55 | 1.33 | 0.22 |
| 03/2012 | 4 | 10 | 1.42 | 0.15 | 10.6 | 1.61 | 1.12 | 0.50 |
| 03/2012 | 5 | 5 | 1.43 | 0.08 | 5.9 | 1.55 | 1.32 | 0.23 |
| 03/2012 | 6 | 5 | 1.28 | 0.30 | 23.4 | 1.73 | 0.93 | 0.81 |
| 03/2012 | 7 | 4 | 1.37 | 0.14 | 10.1 | 1.46 | 1.16 | 0.30 |
| 03/2012 | 8 | 6 | 1.31 | 0.11 | 8.4 | 1.42 | 1.18 | 0.24 |
| 03/2012 | 9 | 5 | 1.43 | 0.10 | 6.9 | 1.51 | 1.26 | 0.25 |
| 03/2012 | 10 | 6 | 1.23 | 0.07 | 5.9 | 1.37 | 1.15 | 0.22 |
| 07/2012 | 11 | 8 | 1.19 | 0.04 | 3.7 | 1.25 | 1.12 | 0.12 |
| 07/2012 | 12 | 8 | 1.42 | 0.14 | 10.2 | 1.64 | 1.20 | 0.44 |
| 07/2012 | 13 | 8 | 1.42 | 0.17 | 12.3 | 1.64 | 1.17 | 0.46 |
| 07/2012 | 14 | 5 | 1.52 | 0.08 | 5.4 | 1.62 | 1.40 | 0.22 |
| 07/2012 | 15 | 7 | 1.44 | 0.09 | 6.4 | 1.59 | 1.34 | 0.26 |
| 07/2012 | 16 | 6 | 1.48 | 0.14 | 9.8 | 1.66 | 1.31 | 0.36 |
| 07/2012 | 17 | 9 | 1.52 | 0.10 | 6.5 | 1.64 | 1.38 | 0.27 |
| 07/2012 | 18 | 9 | 1.29 | 0.09 | 6.7 | 1.41 | 1.18 | 0.23 |
| 07/2012 | 19 | 7 | 1.31 | 0.14 | 10.5 | 1.51 | 1.16 | 0.35 |
| 07/2012 | 20 | 5 | 1.29 | 0.13 | 9.8 | 1.44 | 1.09 | 0.34 |
| 07/2012 | 21 | 8 | 1.38 | 0.13 | 9.1 | 1.63 | 1.23 | 0.40 |
| 07/2012 | 22 | 8 | 1.44 | 0.12 | 8.6 | 1.58 | 1.26 | 0.32 |
| 07/2012 | 23 | 4 | 1.31 | 0.23 | 17.4 | 1.65 | 1.18 | 0.47 |
| 07/2012 | 24 | 8 | 1.43 | 0.15 | 10.4 | 1.74 | 1.29 | 0.45 |
| 07/2012 | 25 | 5 | 1.20 | 0.17 | 13.8 | 1.43 | 1.01 | 0.42 |
| 07/2012 | 26 | 7 | 1.32 | 0.22 | 16.4 | 1.73 | 1.07 | 0.66 |
| 09/2012 | 27 | 5 | 1.21 | 0.15 | 12.2 | 1.44 | 1.06 | 0.37 |
| 09/2012 | 28 | 7 | 1.38 | 0.16 | 11.8 | 1.63 | 1.09 | 0.55 |
| 09/2012 | 29 | 9 | 1.31 | 0.10 | 7.4 | 1.52 | 1.19 | 0.33 |
| 09/2012 | 30 | 8 | 1.23 | 0.08 | 6.8 | 1.38 | 1.10 | 0.28 |
| 09/2012 | 31 | 6 | 1.45 | 0.06 | 4.0 | 1.54 | 1.39 | 0.16 |
| 09/2012 | 32 | 5 | 1.47 | 0.13 | 8.8 | 1.70 | 1.40 | 0.30 |
| 09/2012 | 33 | 4 | 1.25 | 0.12 | 9.5 | 1.36 | 1.09 | 0.27 |
| 09/2012 | 34 | 9 | 1.42 | 0.10 | 7.0 | 1.60 | 1.29 | 0.31 |
| 09/2012 | 35 | 8 | 1.39 | 0.08 | 5.4 | 1.54 | 1.27 | 0.27 |
| 09/2012 | 36 | 7 | 1.39 | 0.15 | 11.2 | 1.64 | 1.14 | 0.50 |
| 09/2012 | 37 | 5 | 1.27 | 0.12 | 9.1 | 1.38 | 1.10 | 0.29 |
| 09/2012 | 38 | 6 | 1.33 | 0.12 | 9.3 | 1.46 | 1.10 | 0.35 |
| 09/2012 | 39 | 6 | 1.37 | 0.06 | 4.6 | 1.42 | 1.26 | 0.17 |
| 09/2012 | 40 | 7 | 1.40 | 0.09 | 6.6 | 1.52 | 1.25 | 0.27 |
| 09/2012 | 41 | 7 | 1.59 | 0.29 | 18.3 | 2.09 | 1.30 | 0.79 |
| 09/2012 | 42 | 4 | 1.42 | 0.07 | 4.9 | 1.50 | 1.34 | 0.16 |
| 09/2012 | 43 | 8 | 1.21 | 0.13 | 10.5 | 1.38 | 1.06 | 0.33 |
| 09/2012 | 44 | 9 | 1.30 | 0.16 | 12.2 | 1.56 | 1.10 | 0.46 |
| | **average** | **6.8** | **1.36** | **0.13** | **9 (±4)** | **1.55** | **1.20** | **0.35** |