# Peer review of "Mn/Ca intra- and inter-test variability in the benthic foraminifer *Ammonia tepida"

_Biogeosciences, 2017_

## Referee Comment (RC1) · Anonymous Referee #1 · 23 Aug 2017

Review of "Mn/Ca intra-test variability in the benthic foraminfer Ammonia tepida" by Petersen et al., submitted to Biogeosciences, Aug., 2017.

These authors present new laser ablation ICP-MS Mn/Ca (Mg/Ca and Sr/Ca) data from individual chambers of a benthic foraminer species taken from the upper sediment of a non-freshwater lake. These data are analyzed with respect to the potential use of Mn/Ca as a proxy for bottom water oxic conditions. The conclusion is that while there may be systematic variability, deconvoluting the three possible sources of non-ontogenetic variability (change in environment, movement of foram and timing of chamber formation) makes such data prohibitively complex. The ms. is very well written and illustrated well. Details of the methodology and results are very good. The conclusions reached are generally supported; in fact, the main criticism I have is that

these data were not explored further (see my comments below). Kudos to the authors for a job well-done.

Major Questions:

1. why not further explore the data? The Mg/Ca and Sr/Ca are only very briefly mentioned. I understand the authors have a story about Mn/Ca and redox to discuss, and I appreciate keeping this story clear. However, their data shows a very large variability in Mg/Ca that does indeed co-vary with Mn/Ca (I calculate a r2 of 0.6). Does temperature vary this much in the lake? Can this help explain Mn/Ca variability (e.g., through a Q10 type foram response)?

2. Following from point 1, it would be useful to have better context for the life environment of these forams. At least there should be a location map where the samples were taken, and some idea of salinity and temperature. Other details, such as organic loading and bioactivity (even human activity) would be useful too. The authors suggest all these data exist - can they plot some of them over the time of interest? Best of all, of course, would be some record of bottom water redox condition over the time of interest. Does any such record exist?

3. More on data comparisons: Can the authors plot the variability of chambers that might be considered time-equivalent? That is, the ultimate chamber of each (presumably live) specimen should be time-equivalent. Can all these be compared? Then similarly for all the second chambers (assuming all the individuals grew similarly, which may be a false assumption, I know.) I did not see such a plot, and could not find the necessary data to generate one myself. This would seem to me to be instructive about variability between individuals that are living in the same chemical environment.

In summary, I would say that this paper is fine as is (with minor revisions), but would be a more substantive contribution with further exploration of the data.

Minor Issues: Cite Froelich et al., 1979, and even add a comment regarding "remnant

Mn peaks" and moving fronts of redox state. These would certainly pertain in this time-sensitive data.

What does "adequately" mean on L.13 pg. 2?

Could there be any problem with Mn or Ca in CellTracker Green?

If these are living (stained) forams, I am confused how high values on the inner and outer shell can be contamination. Of what?

Does their LOD and LOQ not preclude them from measuring low Mn? e.g., what you expect to find in more oxic conditions?

Their statistical analyses seem very robust, but there is not much take away from it.

The range of data in Fig. 4 is perhaps most interesting; is the minimum the same in all cases? i.e., is there a minimum Mn incorporation in shells regardless of environment/ontogeny?

The section on ontogeny on p.16 shoudl come earlier, and provide information on how fast are chambers grown, what kind of time range does each chamber represents, and if the chambers grow at all times of the year. At least to the ability that the authors can provide this information.

Section 4.2.2. - The use of % variability might be a poor option here, as it depends on the Mn/Ca measured.

Conclusion presented (e.g., L. 15, conclusions): the intra-test variability may be caused by environmental change (Mn front shifting), but wouldn't this be recorded more consistently in all the samples?

―――――――――――――――――――――

---

## Referee Comment (RC2) · Anonymous Referee #2 · 28 Aug 2017

Petersen and co-authors present laser ablation derived Mn/Ca ratios of the benthic foraminifera species Ammonia tepida and propose to use them as a proxy for bottom water oxygenation. During three different months, living specimens were collected at a lake with seasonal changes in the redox status of the upper sediment. The results show a high intra-test variability in foraminiferal Mn/Ca ratios due to ontogenetic trends, seasonal changes in pore water $Mn_{2+}$ concentration and vertical migration of the foraminifera within the sediment. The authors ascribe the highest amount of the intra-test variability to variations in the $Mn_{2+}$ concentration of the pore waters and that differences in calcification histories might explain observed inter-test variability.

The manuscript is generally well written, logically organised and clear. The figures are mostly nice and clear. I think that this work is an interesting and important contribution

and therefore suitable to be published in Biogeosciences. Nevertheless, I would like to see the points below addressed by the authors.

Main points

1) I think it is hard to compare this study with the results from the culture study from Barras et al., since their study is submitted to another journal and the reader has no access to the data which makes it hard to verify the results of Petersen et al. Is there another reference that could help here that is already published?

2) Some important information are missing or are not sufficiently explained: - It would be very helpful to have the lake pore water data for $Mn^{2+}$ and $O_2$ concentrations in actual numbers at least for the months investigated to better compare them with the measured foraminiferal Mn/Ca ratios and to assess the redox conditions in the upper sediment.

- In Section 2.3 you say that different spot sizes were used according to different chamber sizes. Did you do test measurements with different spot sizes (on standards and/or foraminifera tests) to show that the spot size does not affect the analysed Mn/Ca (and other) values? Please specify this.

- In Table 2, information about measured NIST 612 standard data are missing as well as measured Mg/Ca ratios for the USGS MACS-3 standard. I think this is important especially when you correct your analysed Mg/Ca ratios to the USGS MACS-3 standard considering the offset in Mg/Ca ratios for NIST 610 standard between your measurements and those of Jochum et al. 2011.

3) There is a contradiction between Sections 3.3 and 4.2.2 concerning ontogenetic trends. In Section 3.3 (page 10, L14-16), you mention that there are statistical significant ontogenetic trends for the analysed data and in Section 4.2.2 (page 14, L5-7) you suddenly say, there were no systematic ontogenetic trends observed. Please specify explicitly that this is only valid if all your data were combined (like you say on page 10,

L16-17).

4) Your justification for analysing the standards in raster mode (page 7, L1-4) is not entirely correct. How can you keep the depth-related fractionation similar for raster and spot analyses, like you say in L2-3? It is true, that measuring in raster mode minimises the "down-hole" fractionation but the fractionation is most likely still different to spot analyses. Please rephrase this. Further, "down-hole" fractionation is probably negligible in foraminifera tests as the test walls are thin – especially when the test walls were entirely ablated within 10s at the mentioned laser settings.

Minor points

- page 1, L25 "intrinsic" – This term is explained later how it is meant in this context. So, please do not use this word in the abstract.

- page 2, L4-6 "...thereby relating bottom water oxygenation,..." – The word "relating" sounds odd in this context. Please rephrase.

- page 2, L17-18 "... lead to hypoxic BWO conditions..." – I think it is better to say here "hypoxic bottom water conditions" otherwise it is kind of double with "hypoxic bottom water oxygenation conditions".

- page 4, L7-8 Did the Mn/Ca intrinsic variability only relate to changes in seawater Mn2+ concentration in the study from Barras et al.? As mentioned before, the reader has no access to the (yet) unpublished study of Barras and co-authors, hence it is difficult to see which factors contribute to the intrinsic variations in Mn/Ca ratios of A. tepida. Please give some more details on this study or state the main/controlling factors for this variability. Is it the Mn2+ concentration as briefly mentioned later?

- page 4, L9-10 Please put "in culturing experiments" in parentheses otherwise it is confusing if your study is based on culturing experiments or on field samples.

- page 4, L10-12 "...species complex..." Correct word? Species group?

- page 4, L17-19 "...represent a suitable context.." – better "suitable location"

- page 4, L19-21 The sentence reads awkward, please rephrase. Maybe: "However, one complicating factor is that it has recently been shown that the activity of cable bacteria strongly influences the seasonal pattern of sediment geochemical cycles in Lake Grevelingen."

- page 7, L17-18 Please insert "the" between "..processed with" and "GLITTER software".

- page 10, L1-2 "...non-parametric test..." – Which one? Please name (again).

- page 10, L15-16 "... there was a slight, significant trend..." - Please delete "slight". If the trend is statistically significant, the word "slight" does not fit.

- page 12, L18-20 Please reference Fig. 3 here instead of Fig. 2 as the variability is better seen in Fig. 3.

- page 13, L10-13 "...unlike Sr, Mg is strongly discriminated against in the calcifying fluid." – What does that mean for the analysed ratios? Please explain briefly.

- page 16, L11-16 As mentioned above, please give the pore water $Mn^{2+}$ concentrations at least for the three months investigated, better for the entire year 2012, if the data were monitored, to better follow your interpretations.

- page 17, L4-8 Please spell OPD out as this is only used twice in the entire manuscript and I really had to look for the explanation.

- page 18, L7-9 "...although no systematic ontogenetic trends could be identified." – Please insert here that this is only the case if the entire dataset is considered. Otherwise it will be a contradiction to your result section as mentioned above in the main points.

References – I am sorry but it is awful to read the reference list. Could you please use indentation "hanging" to make it easier to read?

Figure 2, SEM image – This is actually Fig. 1. Out of curiosity, why does this test have 11 ablation holes if 10 spots were analysed at maximum (according to Fig. 2, plots)?

Figure 2, average Mn/Ca ratios per chamber and specimen – I really like this plot but each panel is very small which makes it hard to actually read the numbers. So, is there another way to show the data? Or at least, please lose the gray background and the grid lines and make each individual panel a bit bigger.

Scientific significance: good Scientific quality: good to fair Presentation quality: good

---

## Author Comment (AC1) · 10 Nov 2017

Reply to Referee comment 1 (RC1):

Below we have copied the referees' comments one at the time and indicate how we have addressed them. Our reply contains two figures as well as an attachment with both RCs' comments and replies and a typed manuscript, which is accompanied by five figures, three tables, one appendix and one supplementary material.

"Review of "Mn/Ca intra-test variability in the benthic foraminfer Ammonia tepida" by Petersen et al., submitted to Biogeosciences, Aug., 2017. These authors present new laser ablation ICP-MS Mn/Ca (Mg/Ca and Sr/Ca) data from individual chambers of a benthic foraminer species taken from the upper sediment of a non-freshwater lake.

[Figure]

These data are analyzed with respect to the potential use of Mn/Ca as a proxy for bottom water oxic conditions. The conclusion is that while there may be systematic variability, deconvoluting the three possible sources of non-ontogenetic variability (change in environment, movement of foram and timing of chamber formation) makes such data prohibitively complex. The ms. is very well written and illustrated well. Details of the methodology and results are very good. The conclusions reached are generally supported; in fact, the main criticism I have is that these data were not explored further (see my comments below). Kudos to the authors for a job well-done."

Reply: We thank the referee for the positive comments

Major Questions:

"1. why not further explore the data? The Mg/Ca and Sr/Ca are only very briefly mentioned. I understand the authors have a story about Mn/Ca and redox to discuss, and I appreciate keeping this story clear. However, their data shows a very large variability in Mg/Ca that does indeed co-vary with Mn/Ca (I calculate a r2 of 0.6). Does temperature vary this much in the lake? Can this help explain Mn/Ca variability (e.g., through a Q10 type foram response)?"

Reply: The referee raises some relevant questions about the data of Mg/Ca and possible relations to temperature variability. As we explain below, the mechanisms behind the Mg/Ca ratios are highly complex, and their discussion would indeed take the attention away from the story line in the manuscript. Regarding the bottom water temperature, we know that it varies from about 5°C in winter to about 18°C in summer at the sampling site in Lake Grevelingen (Fig. 1 of this author's response; from Hagens et al., 2015, Biogeosciences).

Average Mg/Ca is 2.4 mmol/mol for March, 3.7 mmol/mol for July and 2.2 mmol/mol for September. Only the difference in Mg/Ca between the latter two months is significant (at a 95% level). According to the calibration of De Nooijer et al. (2014a) for A. tepida, these values would correspond to high absolute temperatures (for all months

well above the 18°C measured in summer) and a maximal temperature difference of approximately 4.3°C. However, a higher temperature in July than in September does not entirely correspond with the observations, which show high temperatures until the end of September. Summarising, at our study site, the relation between Mg/Ca and temperature appears to be complex. It is therefore unlikely that specimens which have both elevated Mn/Ca and Mg/Ca reflect a Q10 type response to increased temperature.

There is indeed a positive correlation between Mg/Ca and Mn/Ca, but only for the mean values per specimen ($R^2$ = 0.6). For the parameters of intra-test variability (range and % RSD) the correlation coefficients are much lower (R2 = 0.31 for the range and R2 = 0.15 for the RSD). Also the individual measurements show a much weaker correlation (R2 = 0.32), which is largely based on about 14 of the 297 measurements (the points with Mn/Ca > 0.4 mmol/mol in Fig. 2 of this author's response).

In the text we suggest that higher intra-test variability tends to lead to a higher average Mn/Ca. For Mn/Ca, maximal intra-test variability was observed in July, possibly because the foraminifera collected in July may still record the winter Mn2+ maximum in some of the chambers. However, also 3 specimens with strongly increased Mn/Ca in a single chamber, which we tentatively interpreted as transport of the foraminifera to deeper sediment layers due to macrofaunal bioturbation, substantially contribute to higher variability in Mn/Ca. It is possible that maximal bioturbation in late spring (after macrofaunal repopulation in early spring) coincides with the period of strong temperature increase, thereby explaining part of the positive correlation between Mg/Ca and Mn/Ca. Another explanation for the positive correlation is the fact that also during summer hypoxia, there is a slight increase in sediment Mn2+ concentrations. In the manuscript we have added more detail to the fact that this latter aspect has been observed in Lake Grevelingen at our sampling site (page 15 line 19-22).

Summarising, it is highly unlikely that average elemental ratios of the measured specimens are representative of the time of sample collection; this makes it challenging to directly relate environmental parameters with shell chemistry. Therefore, we suggest

keeping the focus of our manuscript only on the Mn/Ca variability and the linkage with redox conditions.

"2. Following from point 1, it would be useful to have better context for the life environment of these forams. At least there should be a location map where the samples were taken, and some idea of salinity and temperature. Other details, such as organic loading and bioactivity (even human activity) would be useful too. The authors suggest all these data exist - can they plot some of them over the time of interest? Best of all, of course, would be some record of bottom water redox condition over the time of interest. Does any such record exist?"

Reply: For the location map a supplementary figure has been added to the manuscript (Figure S1). As mentioned before, temperature and salinity are published in Hagens et al. (2015) which is now explicitly mentioned in the manuscript (page 5 line 17/18). Information about organic loading can be obtained from Seitaj et al. (2017). The record of bottom water oxygenation for the sampled station is now added as a figure to the supplementary material (Figure S2).

"3. More on data comparisons: Can the authors plot the variability of chambers that might be considered time-equivalent? That is, the ultimate chamber of each (presumably live) specimen should be time-equivalent. Can all these be compared? Then similarly for all the second chambers (assuming all the individuals grew similarly, which may be a false assumption, I know.) I did not see such a plot, and could not find the necessary data to generate one myself. This would seem to me to be instructive about variability between individuals that are living in the same chemical environment."

Reply: The plot that the referee asks for was already included in the manuscript as Fig. 5. For all specimens from one sample (and for all specimens combined) all the penultimate, antepenultimate, etc. chambers are plotted together. However, as already suggested by the referee, it is not certain that all penultimate chambers from different specimens of the same sample calcified at the same time (and hence, under

the same conditions). Nevertheless, we have added some discussion (page 17, line 9-13), explaining that this figure indeed reflects temporal variability of Mn/Ca in relation to environmental changes, but that the different calcification histories of individual specimens add uncertainty to this interpretation.

In fact, we performed such a comparison on the basis of a substantially larger data set than the one presented in this manuscript, to test whether specimens from the same sample produced their chambers simultaneously (which did not appear to be the case). In order to keep our manuscript focused, we did not add this analysis to our manuscript. Instead, the relation between Mn/Ca, environmental conditions and timing of calcification will be discussed in detail in a future manuscript.

Minor Issues:

"Cite Froelich et al., 1979, and even add a comment regarding "remnant Mn peaks" and moving fronts of redox state. These would certainly pertain in this time-sensitive data."

Reply: Froelich et al. (1979) is cited on page 2 line 7/8. To address the moving fronts of redox states changes have been made on page 2 line 17-19. Regarding remnant Mn peaks, this was observed by Froelich et al., however, in the Lake Grevelingen data the Mn2+ (and also solid phase Mn) show no such pattern, so we refrain from adding too much detail in the introduction.

"What does "adequately" mean on L.13 pg. 2?"

Reply: This comment refers to page 3 line 12. Indeed, the word does not add further information here, so it was deleted.

"Could there be any problem with Mn or Ca in CellTracker Green?"

Reply: Samples treated with CellTracker Green (5-chloromethylfluorescein diacetate) were incubated for at least 6 h before they were fixed with formalin. Although possible, it is unlikely that foraminifera calcified during the reaction period, and it is highly unlikely

that more than one chamber would be added during this time interval. Since we did not analyse the geochemical composition of the last chamber, we are sure not to have sampled a chamber that was calcified during incubation with CTG.

Another possible influence of CTG on the test could be adsorption effects on the outside of chamber walls. However, Bernhard et al. (2006) point out that CTG will not leak out of the cell via ion channels in the cell membrane once it is incorporated inside the cell. Therefore, it is unlikely that there is an interference with the calcite from this organic compound (long) after the forams are stained. Moreover, the laser ablation signal from the outside of chamber walls is omitted from further data treatment.

"If these are living (stained) forams, I am confused how high values on the inner and outer shell can be contamination. Of what?"

Reply: It is considered that sediment particles are attached to the surface of tests of living benthic foraminifera even after the cleaning procedure (Koho et al., 2015, 2017).

"Does their LOD and LOQ not preclude them from measuring low Mn? e.g., what you expect to find in more oxic conditions?"

Reply: We expect calcification to normally occur at the sediment-water interface where oxic conditions prevail throughout most of the year. Therefore, we consider the Mn/Ca measured for most of the chambers, with a large majority of values (207 out of 298 measurements) between 0.05 and 0.2 mmol/mol, Fig. 2, to be representative for oxic conditions.

However, we agree with the referee that we cannot exclude that some of the measurements below the LOQ correspond to very low Mn/Ca values calcified in oxic conditions. Fortunately, only 4 of the 44 analysed specimens presented more than 2 chambers with values below the LOQ. Since on average, 7.7 chambers were measured per specimen, the impact on the average value and on the intra-test variability for the individual would be small, even more so, because a large majority of the measurements below

the LOQ is found in specimens with systematically low Mn/Ca values (and a low Mn/Ca intra-test variability).

"The range of data in Fig. 4 is perhaps most interesting; is the minimum the same in all cases? i.e., is there a minimum Mn incorporation in shells regardless of environment/ontogeny?"

Reply: Minimum Mn/Ca values are listed in Tab. A.1. On average, this was 0.08 mmol/mol, but depending on the specimen it was as low as 0.03 mmol/mol or as high as 0.19 mmol/mol. Concerning minimum Mn incorporation regardless of environment, it is probably best to refer to culturing studies because it is the only way to keep Mn2+ concentrations in the solution constantly low.

"The section on ontogeny on p.16 should come earlier, and provide information on how fast are chambers grown, what kind of time range does each chamber represents, and if the chambers grow at all times of the year. At least to the ability that the authors can provide this information."

Reply: Although we understand the referee's suggestion that this important information should be stated earlier in the text, we think that it is at its right place here in the discussion, where temporal factors are discussed. Unfortunately there is not much more information available to our knowledge, so that we cannot provide more details.

"Section 4.2.2. - The use of % variability might be a poor option here, as it depends on the Mn/Ca measured."

Reply: We think that the use of % RSD (i.e., the standard deviation normalized to the average value) is correct here because it is exactly our intention to compare the extent of variability between different absolute values. The % RSD is the most accepted measure which allows to do so.

"Conclusion presented (e.g., L. 15, conclusions): the intra-test variability may be caused by environmental change (Mn front shifting), but wouldn't this be recorded more

consistently in all the samples?"

Reply: We think it is not consistently recorded because of the "different timing of calcification" of individual specimens, as stated in the conclusion, page 19 line 6-8.

Please also note the supplement to this comment:
https://www.biogeosciences-discuss.net/bg-2017-273/bg-2017-273-AC1-supplement.pdf
* * *
[Figure]

**Fig. 1.** Water column parameters (sampling station for this study at 23 m water depth). A: temperature [°C]. B: salinity. C: density anomaly [kg/m3]. D: oxygen conc. [$\mu$mol/L]. From Hagens et al. (2015).

[Figure]

Mn/Ca as a function of Mg/Ca plot with axes Mn/Ca [mmol/mol] (y-axis) and Mg/Ca [mmol/mol] (x-axis).

$$y = 0.098 + 0.027 \cdot x, \; R^2 = 0.324, \; p = 6.86e\text{-}27$$

**Fig. 2.** Mn/Ca as a function of Mg/Ca for all single chamber measurements (n=297).

**Supplement:**

Dear editor,

Please find enclosed our final author comments. We thank the referees for their constructive comments which we address here and also enclose the revised manuscript as a supplement with changes tracked in red. We would like to point out that, in order to better reflect the contents, we decided to slightly change the title to "Mn/Ca intra- and inter-test variability in the benthic foraminifer *Ammonia tepida*". This is due to the fact that the manuscript includes many aspects of comparisons between specimens of different samplings.

Thank you for considering our manuscript for publication in Biogeosciences.

Kind regards and on behalf of all co-authors,

Jassin Petersen

RC1: changes in red

"Review of "Mn/Ca intra-test variability in the benthic foraminfer Ammonia tepida" by Petersen et al., submitted to Biogeosciences, Aug., 2017. These authors present new laser ablation ICP-MS Mn/Ca (Mg/Ca and Sr/Ca) data from individual chambers of a benthic foraminer species taken from the upper sediment of a non-freshwater lake. These data are analyzed with respect to the potential use of Mn/Ca as a proxy for bottom water oxic conditions. The conclusion is that while there may be systematic variability, deconvoluting the three possible sources of non-ontogenetic variability (change in environment, movement of foram and timing of chamber formation) makes such data prohibitively complex. The ms. is very well written and illustrated well. Details of the methodology and results are very good. The conclusions reached are generally supported; in fact, the main criticism I have is that these data were not explored further (see my comments below). Kudos to the authors for a job well-done."

Reply: We thank the referee for the positive comments

Major Questions:

"1. why not further explore the data? The Mg/Ca and Sr/Ca are only very briefly mentioned. I understand the authors have a story about Mn/Ca and redox to discuss, and I appreciate keeping this story clear. However, their data shows a very large variability in Mg/Ca that does indeed co-vary with Mn/Ca (I calculate a r2 of 0.6). Does temperature vary this much in the lake? Can this help explain Mn/Ca variability (e.g., through a Q10 type foram response)?"

Reply: The referee raises some relevant questions about the data of Mg/Ca and possible relations to temperature variability. As we explain below, the mechanisms behind the Mg/Ca ratios are highly complex, and their discussion would indeed take the attention away from the story line in the manuscript.

Regarding the bottom water temperature, we know that it varies from about 5°C in winter to about 18°C in summer at the sampling site in Lake Grevelingen (Fig. 1 of this author's response; from Hagens et al., 2015, Biogeosciences).

[Figure]

Figure 1. Water column parameters for sampling campaigns in 2012, linearly interpolated in space and time. Sampling station for this study is situated at 23 m water depth. A: temperature [°C]. B: salinity. C: density anomaly [kg/m³]. D: oxygen concentration [µmol/L]. Modified from Hagens et al. (2015).

Average Mg/Ca is 2.4 mmol/mol for March, 3.7 mmol/mol for July and 2.2 mmol/mol for September. Only the difference in Mg/Ca between the latter two months is significant (at a 95% level). According to the calibration of De Nooijer et al. (2014a) for *A. tepida*, these values would correspond to high absolute temperatures (for all months well above the 18°C measured in summer) and a maximal temperature difference of approximately 4.3°C. However, a higher temperature in July than in September does not entirely correspond with the observations, which show high temperatures until the end of September. Summarising, at our study site, the relation between Mg/Ca and temperature appears to be complex. It is therefore unlikely that specimens which have both elevated Mn/Ca and Mg/Ca reflect a Q10 type response to increased temperature.

There is indeed a positive correlation between Mg/Ca and Mn/Ca, but only for the mean values per specimen (R² = 0.6). For the parameters of intra-test variability (range and % RSD) the correlation coefficients are much lower ($R^2$ = 0.31 for the range and $R^2$ = 0.15 for the RSD). Also the individual measurements show a much weaker correlation ($R^2$ = 0.32), which is largely based on about 14 of the 297 measurements (the points with Mn/Ca > 0.4 mmol/mol in Fig. 2 of this author's response).

[Figure]

Figure 2. Mn/Ca as a function of Mg/Ca for all single chamber measurements (n=297).

In the text we suggest that higher intra-test variability tends to lead to a higher average Mn/Ca. For Mn/Ca, maximal intra-test variability was observed in July, possibly because the foraminifera collected in July may still record the winter $Mn^{2+}$ maximum in some of the chambers. However, also 3 specimens with strongly increased Mn/Ca in a single chamber, which we tentatively interpreted as transport of the foraminifera to deeper sediment layers due to macrofaunal bioturbation, substantially contribute to higher variability in Mn/Ca. It is possible that maximal bioturbation in late spring (after macrofaunal repopulation in early spring) coincides with the period of strong temperature increase, thereby explaining part of the positive correlation between Mg/Ca and Mn/Ca. Another explanation for the positive correlation is the fact that also during summer hypoxia, there is a slight increase in sediment $Mn^{2+}$ concentrations. In the manuscript we have added more detail to the fact that this latter aspect has been observed in Lake Grevelingen at our sampling site (page 15 line 19-22). Summarising, it is highly unlikely that average elemental ratios of the measured specimens are representative of the time of sample collection; this makes it challenging to directly relate environmental parameters with shell chemistry. Therefore, we suggest keeping the focus of our manuscript only on the Mn/Ca variability and the linkage with redox conditions.

"2. Following from point 1, it would be useful to have better context for the life environment of these forams. At least there should be a location map where the samples were taken, and some idea of salinity and temperature. Other details, such as organic loading and bioactivity (even human activity) would be useful too. The authors suggest all these data exist - can they plot some of them over the time of interest? Best of all, of course, would be some record of bottom water redox condition over the time of interest. Does any such record exist?"

Reply: For the location map a supplementary figure has been added to the manuscript (Figure S1). As mentioned before, temperature and salinity are published in Hagens et al. (2015) which is now explicitly mentioned in the manuscript (page 5 line 17/18). Information

about organic loading can be obtained from Seitaj et al. (2017). The record of bottom water oxygenation for the sampled station is now added as a figure to the supplementary material (Figure S2).

"3. More on data comparisons: Can the authors plot the variability of chambers that might be considered time-equivalent? That is, the ultimate chamber of each (presumably live) specimen should be time-equivalent. Can all these be compared? Then similarly for all the second chambers (assuming all the individuals grew similarly, which may be a false assumption, I know.) I did not see such a plot, and could not find the necessary data to generate one myself. This would seem to me to be instructive about variability between individuals that are living in the same chemical environment."

Reply: The plot that the referee asks for was already included in the manuscript as Fig. 5. For all specimens from one sample (and for all specimens combined) all the penultimate, antepenultimate, etc. chambers are plotted together. However, as already suggested by the referee, it is not certain that all penultimate chambers from different specimens of the same sample calcified at the same time (and hence, under the same conditions). Nevertheless, we have added some discussion (page 17, line 9-13), explaining that this figure indeed reflects temporal variability of Mn/Ca in relation to environmental changes, but that the different calcification histories of individual specimens add uncertainty to this interpretation.
In fact, we performed such a comparison on the basis of a substantially larger data set than the one presented in this manuscript, to test whether specimens from the same sample produced their chambers simultaneously (which did not appear to be the case). In order to keep our manuscript focused, we did not add this analysis to our manuscript. Instead, the relation between Mn/Ca, environmental conditions and timing of calcification will be discussed in detail in a future manuscript.

"Minor Issues: Cite Froelich et al., 1979, and even add a comment regarding "remnant Mn peaks" and moving fronts of redox state. These would certainly pertain in this time-sensitive data."

Reply: Froelich et al. (1979) is cited on page 2 line 7/8. To address the moving fronts of redox states changes have been made on page 2 line 17-19. Regarding remnant Mn peaks, this was observed by Froelich et al., however, in the Lake Grevelingen data the $Mn^{2+}$ (and also solid phase Mn) show no such pattern, so we refrain from adding too much detail in the introduction.

"What does "adequately" mean on L.13 pg. 2?"

Reply: This comment refers to page 3 line 12. Indeed, the word does not add further information here, so it was deleted.

"Could there be any problem with Mn or Ca in CellTracker Green?"

Reply: Samples treated with CellTracker Green (5-chloromethylfluorescein diacetate) were incubated for at least 6 h before they were fixed with formalin. Although possible, it is unlikely that foraminifera calcified during the reaction period, and it is highly unlikely that more than one chamber would be added during this time interval. Since we did not analyse the geochemical composition of the last chamber, we are sure not to have sampled a chamber that was calcified during incubation with CTG.

Another possible influence of CTG on the test could be adsorption effects on the outside of chamber walls. However, Bernhard et al. (2006) point out that CTG will not leak out of the cell via ion channels in the cell membrane once it is incorporated inside the cell. Therefore, it is unlikely that there is an interference with the calcite from this organic compound (long) after the forams are stained. Moreover, the laser ablation signal from the outside of chamber walls is omitted from further data treatment.

"If these are living (stained) forams, I am confused how high values on the inner and outer shell can be contamination. Of what?"

Reply: It is considered that sediment particles are attached to the surface of tests of living benthic foraminifera even after the cleaning procedure (Koho et al., 2015, 2017).

"Does their LOD and LOQ not preclude them from measuring low Mn? e.g., what you expect to find in more oxic conditions?"

Reply: We expect calcification to normally occur at the sediment-water interface where oxic conditions prevail throughout most of the year. Therefore, we consider the Mn/Ca measured for most of the chambers, with a large majority of values (207 out of 298 measurements) between 0.05 and 0.2 mmol/mol, Fig. 2, to be representative for oxic conditions. However, we agree with the referee that we cannot exclude that some of the measurements below the LOQ correspond to very low Mn/Ca values calcified in oxic conditions. Fortunately, only 4 of the 44 analysed specimens presented more than 2 chambers with values below the LOQ. Since on average, 7.7 chambers were measured per specimen, the impact on the average value and on the intra-test variability for the individual would be small, even more so, because a large majority of the measurements below the LOQ is found in specimens with systematically low Mn/Ca values (and a low Mn/Ca intra-test variability).

"The range of data in Fig. 4 is perhaps most interesting; is the minimum the same in all cases? i.e., is there a minimum Mn incorporation in shells regardless of environment/ ontogeny?"

Reply: Minimum Mn/Ca values are listed in Tab. A.1. On average, this was 0.08 mmol/mol, but depending on the specimen it was as low as 0.03 mmol/mol or as high as 0.19 mmol/mol. Concerning minimum Mn incorporation regardless of environment, it is probably best to refer to culturing studies because it is the only way to keep $Mn^{2+}$ concentrations in the solution constantly low.

"The section on ontogeny on p.16 should come earlier, and provide information on how fast are chambers grown, what kind of time range does each chamber represents, and if the chambers grow at all times of the year. At least to the ability that the authors can provide this information."

Reply: Although we understand the referee's suggestion that this important information should be stated earlier in the text, we think that it is at its right place here in the discussion, where temporal factors are discussed. Unfortunately there is not much more information available to our knowledge, so that we cannot provide more details.

"Section 4.2.2. - The use of % variability might be a poor option here, as it depends on the Mn/Ca measured."

Reply: We think that the use of % RSD (i.e., the standard deviation normalized to the average value) is correct here because it is exactly our intention to compare the extent of variability between different absolute values. The % RSD is the most accepted measure which allows to do so.

"Conclusion presented (e.g., L. 15, conclusions): the intra-test variability may be caused by environmental change (Mn front shifting), but wouldn't this be recorded more consistently in all the samples?"

Reply: We think it is not consistently recorded because of the "different timing of calcification" of individual specimens, as stated in the conclusion, page 19 line 6-8.

RC2: changes in dark red

"Petersen and co-authors present laser ablation derived Mn/Ca ratios of the benthic foraminifera species Ammonia tepida and propose to use them as a proxy for bottom water oxygenation. During three different months, living specimens were collected at a lake with seasonal changes in the redox status of the upper sediment. The results show a high intra-test variability in foraminiferal Mn/Ca ratios due to ontogenetic trends, seasonal changes in pore water $Mn^{2+}$ concentration and vertical migration of the foraminifera within the sediment. The authors ascribe the highest amount of the intra-test variability to variations in the $Mn^{2+}$ concentration of the pore waters and that differences in calcification histories might explain observed inter-test variability. The manuscript is generally well written, logically organised and clear. The figures are mostly nice and clear. I think that this work is an interesting and important contribution and therefore suitable to be published in Biogeosciences. Nevertheless, I would like to see the points below addressed by the authors."

Reply: We thank the referee for the positive comments.

Main points

"1) I think it is hard to compare this study with the results from the culture study from Barras et al., since their study is submitted to another journal and the reader has no access to the data which makes it hard to verify the results of Petersen et al. Is there another reference that could help here that is already published?"

Reply: Indeed, there is another published culturing study of *A. tepida* with seawater $Mn^{2+}$ as the controlling factor (Munsel et al., 2010). However, this study used relatively low concentrations of $Mn^{2+}$ (11-220 nmol/L), compared to those used by Barras et al. (2-595 µmol/L) and found in the pore waters of Lake Grevelingen at our station (up to 310 µmol/L). Moreover, the study of Munsel et al. (2010) was performed under oxic conditions (whereas Barras et al. maintained hypoxic conditions), which could lead to oxide and hydroxide formation (as mentioned by Munsel et al.). For these reasons we cannot compare our Mn/Ca intra-test variability only with the data of Munsel et al. (2010). We have added a sentence explaining this in the new version of the manuscript (page 4, line 5-8). Regarding the study of Barras et al., this manuscript is still under review but following the editor's suggestions it should be accepted after revision. However, we made sure that all relevant information of this paper is given in detail in our text. More specifically, we modified the introduction substantially to give the reader more detailed information (page 4, line 8-11).

"2) Some important information are missing or are not sufficiently explained: - It would be very helpful to have the lake pore water data for Mn2+ and O2 concentrations in actual numbers at least for the months investigated to better compare them with the measured foraminiferal Mn/Ca ratios and to assess the redox conditions in the upper sediment."

Reply: This was partly also suggested by referee 1, so we added supplementary material including a figure of $O_2$ concentration in Lake Grevelingen for 2012 (Fig. S2), and the $Mn^{2+}$ pore water profiles for the three investigated months (Fig. S3).

"- In Section 2.3 you say that different spot sizes were used according to different chamber sizes. Did you do test measurements with different spot sizes (on standards and/or foraminifera tests) to show that the spot size does not affect the analysed Mn/Ca (and other) values? Please specify this."

Reply: The spot sizes of 40-85 µm in diameter used in the course of this study can be compared to the depths of the laser ablation drilling holes to evaluate possible influences on results (according to Eggins et al., 1998, Applied Surface Science, it is this aspect ratio between depth and diameter that determines the fractionation at constant laser energy). In our case the depth is constrained by the thickness of the chambers (probably not exceeding 30 µm and mostly more close to 10 µm for our specimens of *Ammonia tepida*). Given this shallow depth in comparison to the diameter we do not expect the spot size to have a significant impact on the results.

"In Table 2, information about measured NIST 612 standard data are missing as well as measured Mg/Ca ratios for the USGS MACS-3 standard. I think this is important especially when you correct your analysed Mg/Ca ratios to the USGS MACS-3 standard considering the offset in Mg/Ca ratios for NIST 610 standard between your measurements and those of Jochum et al. 2011."

Reply: Table 2 does not include data for NIST 612 because we used this reference material as calibration standard for all other measurements for Mn/Ca and Sr/Ca. Therefore, this material cannot be used to assess external reproducibility. Similarly, we do not report results of Mg/Ca for USGS MACS-3 in Tab. 2 because this was the calibration standard for Mg/Ca. Regarding the offset for NIST 610 between our measurements and those of Jochum et al. (2011), we have to point out that Mg seems to be subject to larger uncertainties in both carbonates and silicates compared to other elements (Jochum et al., 2012).

"3) There is a contradiction between Sections 3.3 and 4.2.2 concerning ontogenetic trends. In Section 3.3 (page 10, L14-16), you mention that there are statistical significant ontogenetic trends for the analysed data and in Section 4.2.2 (page 14, L5-7) you suddenly say, there were no systematic ontogenetic trends observed. Please specify explicitly that this is only valid if all your data were combined (like you say on page 10, L16-17)."

Reply: We added this information in section 4.2.2 (page 14, line 16/17).

"4) Your justification for analysing the standards in raster mode (page 7, L1-4) is not entirely correct. How can you keep the depth-related fractionation similar for raster and spot analyses, like you say in L2-3? It is true, that measuring in raster mode minimises the "down-hole" fractionation but the fractionation is most likely still different to spot analyses. Please rephrase this. Further, "down-hole" fractionation is probably negligible in foraminifera tests as the test walls are thin – especially when the test walls were entirely ablated within 10s at the mentioned laser settings."

Reply: We have changed the sentences accordingly (page 7, line 8-12).

Minor points
"page 1, L25 "intrinsic" – This term is explained later how it is meant in this context. So, please do not use this word in the abstract."

Reply: The word was removed and replaced by "ontogenetic trends (i.e., size-related effects) and/or other vital effects occurring during calcification".

"page 2, L4-6 ": : :thereby relating bottom water oxygenation,: : :" – The word "relating" sounds odd in this context. Please rephrase."

Reply: The word has been replaced by "coupling".

"page 2, L17-18 ": : : lead to hypoxic BWO conditions: : :" – I think it is better to say here "hypoxic bottom water conditions" otherwise it is kind of double with "hypoxic bottom water oxygenation conditions"."

Reply: Changes have been made.

"page 4, L7-8 Did the Mn/Ca intrinsic variability only relate to changes in seawater Mn2+ concentration in the study from Barras et al.? As mentioned before, the reader has no access to the (yet) unpublished study of Barras and co-authors, hence it is difficult to see which factors contribute to the intrinsic variations in Mn/Ca ratios of A. tepida. Please give some more details on this study or state the main/controlling factors for this variability. Is it the Mn2+ concentration as briefly mentioned later?"

Reply: We added on page 4 line 8-11 that the seawater $Mn^{2+}$ concentration was the controlling parameter in the study of Barras et al. and following the referee's first main point more detail has been added to describe this culturing study.

"page 4, L9-10 Please put "in culturing experiments" in parentheses otherwise it is confusing if your study is based on culturing experiments or on field samples."

Reply: Done (page 4, line 14/15).

"page 4, L10-12 ": : :species complex: : :" Correct word? Species group?"

Reply: Page 4 line 16: Species complex is the correct term in this context describing cryptic species. For *Ammonia tepida* there are several phylotypes present in Europe, with slight morphological differences, making this species complex pseudocryptic. For our study we did not perform genetic tests on the specimens analysed so that we use the term *A. tepida* without specifying the phylotype. However, there is an ongoing study on the genetics of *A. tepida* from Lake Grevelingen.

"page 4, L17-19 ": : :represent a suitable context.." – better "suitable location"."

Reply: Done (Page 4 line 22 and page 5 line 1/2).

"page 4, L19-21 The sentence reads awkward, please rephrase. Maybe: "However, one complicating factor is that it has recently been shown that the activity of cable bacteria strongly influences the seasonal pattern of sediment geochemical cycles in Lake Grevelingen." "

Reply: Changes have been made (page 5 line 2-5).

"page 7, L17-18 Please insert "the" between "..processed with" and "GLITTER software"."

Reply: Done (page 8 line 5).

"page 10, L1-2 ": : :non-parametric test: : :" – Which one? Please name (again)."

Reply: Done (page 10 line 11/12).

"page 10, L15-16 ": : : there was a slight, significant trend: : :" - Please delete "slight". If the trend is statistically significant, the word "slight" does not fit."

Reply: We deleted "slight" before "significant trend". To characterise the slope of this correlation we added "slightly" in front of "increasing values" (page 11 line 5/6).

"page 12, L18-20 Please reference Fig. 3 here instead of Fig. 2 as the variability is better seen in Fig. 3."

Reply: Done (page 13 line 7).

"page 13, L10-13 ": : :unlike Sr, Mg is strongly discriminated against in the calcifying fluid." – What does that mean for the analysed ratios? Please explain briefly."

Reply: Done (page 13 line 21 and page 14 line 1).

"page 16, L11-16 As mentioned above, please give the pore water $Mn^{2+}$ concentrations at least for the three months investigated, better for the entire year 2012, if the data were monitored, to better follow your interpretations."

Reply: Pore water $Mn^{2+}$ profiles for the three months investigated were added as supplementary material (Fig. S3) and a reference to it was made on page 16 line 12.

"page 17, L4-8 Please spell OPD out as this is only used twice in the entire manuscript and I really had to look for the explanation."

Reply: Done (page 18 line 2/3).

"page 18, L7-9 ": : :although no systematic ontogenetic trends could be identified." – Please insert here that this is only the case if the entire dataset is considered. Otherwise it will be a contradiction to your result section as mentioned above in the main points."

Reply: This was added in parentheses (page 19 line 6).

"References – I am sorry but it is awful to read the reference list. Could you please use indentation "hanging" to make it easier to read?"

Reply: Done.

"Figure 2, SEM image – This is actually Fig. 1. Out of curiosity, why does this test have 11 ablation holes if 10 spots were analysed at maximum (according to Fig. 2, plots)?"

Reply: For a few specimens more than 10 spots were analyzed, however, no specimen had more than ten ablation spots fulfilling the selection criteria, this is why in Fig. 2 the x-axis is set to 10.

"Figure 2, average Mn/Ca ratios per chamber and specimen – I really like this plot but each panel is very small which makes it hard to actually read the numbers. So, is there another way to show the data? Or at least, please lose the gray background and the grid lines and make each individual panel a bit bigger."

Reply: We modified Figure 2 according to the referee's suggestions.

[revised manuscript text omitted]
** | **6.8** | **1.36** | **0.13** | **9 (±4)** | **1.55** | **1.20** | **0.35** |

[Figure]

**Figure S1. Location map of Lake Grevelingen and sampling station for the material used in this study. A: map of the Netherlands. B: bathymetry of Lake Grevelingen, the sluice giving access to the North Sea is situated at the most western end of the lake. A and B modified from Hagens et al. (2015). C: satellite image of sampling station Grev-2 (image from Google Earth, date of image 8/7/2013).**

[Figure]

**Figure S2. Oxygen concentration in 2012 for sampled station in Lake Grevelingen (depth=23.1 m, for location see Fig. S1). Dashed line indicates threshold for hypoxic conditions (63 µmol/L). Data from Hagens et al. (2015) and Seitaj et al. (2017).**

[Figure]

**Figure S3. Mn²⁺ pore water concentrations for March (A), July (B) and September (C) 2012 at sampled station in Lake Grevelingen. Grey area indicates sampling interval for specimens of *Ammonia tepida* (0-0.5 cm). Data from Sulu-Gambari et al. (2016a, 2016b).**